# Overlapping speckle correlation algorithm for high-resolution imaging and tracking of objects in unknown scattering media

Yaoyao Shi [1,2,3] ✉, Wei Sheng [1], Yangyang Fu [1] ✉ & Youwen Liu [1] ✉

Optical imaging in scattering media is important to many fields but remains challenging. Recent methods have focused on imaging through thin scattering layers or thicker scattering media with prior knowledge of the sample, but this still limits practical applications. Here, we report an imaging method named 'speckle kinetography' that enables high-resolution imaging in unknown scattering media with thicknesses up to about 6 transport mean free paths. Speckle kinetography non-invasively records a series of incoherent speckle images accompanied by object motion and the inherently retained object information is extracted through an overlapping speckle correlation algorithm to construct the object's autocorrelation for imaging. Under single-colour light-emitting diode, white light, and fluorescence illumination, we experimentally demonstrate 1 μm resolution imaging and tracking of objects moving in scattering samples, while reducing the requirements for prior knowledge. We anticipate this method will enable imaging in currently inaccessible scenarios.

Optical imaging within and through scattering media has important applications in many fields, such as biomedicine[1,2], artificial intelligence[3,4] and astronomy[5]. Scattering media, such as biological tissues, diffuse light into a speckle pattern[6] and scramble the spatial information[7]. Owing to the loss of space invariance[8], imaging is difficult in scattering media. An effective solution is to reestablish the space invariance by extracting the unscattered ballistic photons[9–11] or utilizing the space invariance of speckles[12–14], i.e., the memory effect[15]. However, when the medium thickness is much larger than its scattering mean free path, $l_s$, imaging becomes more difficult since the ballistic photons are exponentially extinguished[13] and the memory effect range drops quickly to the wavelength scale[16,17]. Nevertheless, it is a rewarding challenge since thick scattering media are common in practical scenarios, such as skulls, clouds and thick ice. To address this challenge, great attention has been given to the input–output relations of a scattering medium[17–24]. Based on this, various approaches, such as wavefront shaping[18–20], transmission matrix[21], deep learning[17] and time-of-flight diffuse optical

tomography[22–24], have allowed focusing[18–20] or millimetre[17] and centimetre[22–24] resolution imaging in or through thick scattering media with necessary prior knowledge about the medium.

Here, we pursue a general and fundamental approach for imaging within unknown scattering media while achieving high-resolution, being noninvasive and imaging as deep as possible. Current approaches based on the input–output relation in scattering media have difficulty achieving this challenging goal because they are essentially medium-dependent. Instead, an alternative strategy is to determine the inherent relations between the hidden objects and speckles. Under this strategy, spatial speckle intensity correlation approaches have recently been proposed[25–27]. Although submillimetre resolution imaging of moving objects through unknown heavily scattering media has been achieved, applying these approaches in practical scenarios is still challenging, since they introduce extra problems caused by the indispensable requirements of a motion trajectory across a whole plane, prior knowledge regarding the hidden object's positions and a specific coherent light source.

[1]College of Physics, Nanjing University of Aeronautics and Astronautics, Nanjing 210016, China. [2]College of Astronautics, Nanjing University of Aeronautics and Astronautics, Nanjing 210016, China. [3]Key Laboratory of Radar Imaging and Microwave Photonics, Ministry of Education, Nanjing University of Aeronautics and Astronautics, Nanjing 210016, China. ✉e-mail: syy411@nuaa.edu.cn; yyfu@nuaa.edu.cn; ywliu@nuaa.edu.cn

In this work, we present an imaging method, named speckle kinetography, that shows advantages in imaging within and through scattering media. Based on the meaning of autocorrelation[8], speckle kinetography utilizes the object's relative motion to construct the object's autocorrelation for imaging. The key information for the autocorrelation construction is the overlap and relative position between the object's motion states (Fig. 1a). We show that even though the object is obscured by a scattering medium, the key information is still retained in the incoherent speckles and can be extracted via an overlapping speckle correlation algorithm to accomplish speckle kinetography. Under a simple and speckle pattern-sized motion trajectory (Fig. 1b), a few noninvasive detections of the speckle images are sufficient for imaging. Incoherent illumination, which provides the benefits of abundant information, stable speckles, high resolution and easily accessible sources, can be either narrowband or broadband. Speckle kinetography does not need any prior knowledge and is suitable for unknown media with thicknesses up to about 6 transport mean free paths[14], $l_t$.

## Results

### Overlapping speckle correlations

As shown in Fig. 1, an object with an intensity transmittance of $O(x, y)$ is embedded in a scattering medium that is illuminated via an incoherent light source. The light intensity distribution on the object plane $(x, y)$ is $I_S(x, y)$, which can be decomposed as multiple point light sources[8]. Photons emitted from each point source undergo random paths as they travel through the medium. On the back surface $(\xi, \eta)$ of the medium, some of the photons meet again; only the photons with optical path differences smaller than the coherence length interfere with each other and form a unique speckle $s(\xi, \eta; x, y)$ that is an impulse response. The speckle formed by the object is the incoherent superposition of all impulse responses from point sources

within $O(x, y)I_S(x, y)$. The photons that failed to interfere constitute a slowly varying envelope. Finally, we can see a speckle distributed on an envelope.

A simple imaging system consisting of a lens and a monochrome camera is used to record a series of magnified speckle images $I_1, ..., I_N$ in succession (Fig. 1b). During the detection, the object moves relative to the medium that remains static and uniform in the region containing it. Then, the recorded speckle images are numerically separated into spatially normalized speckles $S_1, ..., S_N$ and slowly varying envelopes $E_1, ..., E_N$ ('Methods' and Supplementary Note 11). Two arbitrary speckles are expressed as follows:

$$S_i(\xi, \eta) = \iint O(x, y)I_S(x, y)s(\xi, \eta; x, y)dxdy \tag{1}$$

$$S_j(\xi, \eta; \Delta x_{ij}, \Delta y_{ij}) = \iint O(x - \Delta x_{ij}, y - \Delta y_{ij})I_S(x, y)s(\xi, \eta; x, y)dxdy \tag{2}$$

where $(\Delta x_{ij}, \Delta y_{ij})$ represents the position of the object at the $i$th detection relative to the $j$th detection. When $i = j$, $\Delta x_{ij} = \Delta y_{ij} = 0$.

Owing to low coherence and strong scattering, the impulse responses are sparse. Thus, we assume that the impulse responses from different point sources with relative distances longer than the spatial coherent length do not overlap with each other. Under this assumption, the overlapping speckle between these two speckles is extracted through a Hadamard product as follows:

$$M_{ij}(\xi, \eta; \Delta x_{ij}, \Delta y_{ij}) = S_i(\xi, \eta)S_j(\xi, \eta; \Delta x_{ij}, \Delta y_{ij})$$
$$= \iint O(x, y)O(x - \Delta x_{ij}, y - \Delta y_{ij})I_S^2(x, y)s^2(\xi, \eta; x, y)dxdy \tag{3}$$

The result only retains impulse responses from the point sources within the overlapping region of the object before and after the motion

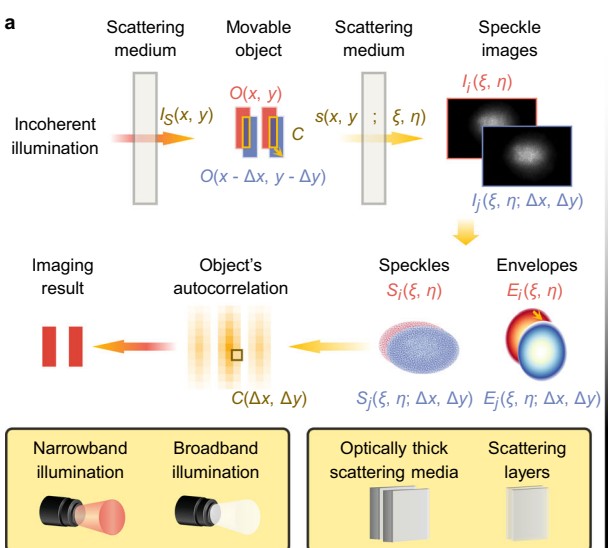

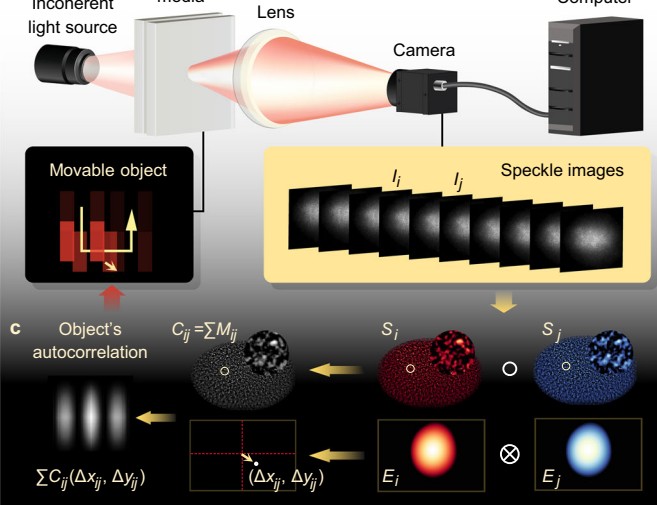

**Fig. 1 | Schematic of the principle, apparatus and computational model for speckle kinetography. a** Principle. Under incoherent illumination, the intensity distribution on object plane is $I_S(x, y)$. Every point source forms an impulse response $s(\xi, \eta; x, y)$ on the back surface of scattering medium. The recorded speckle image $I$ is the superposition of the impulse responses generated within the object. The information of overlap $C$ and relative position $(\Delta x, \Delta y)$ between any two motion states of an object $O(x, y)$ and $O(x-\Delta x, y-\Delta y)$ is retained in the speckle images. The speckle $S_i(\xi, \eta)$ and the envelope $E_i(\xi, \eta)$ in red are respectively high-pass and low-pass filtered from the speckle image $I_i(\xi, \eta)$. Similarly, $S_j(\xi, \eta; \Delta x, \Delta y)$ and $E_j(\xi, \eta; \Delta x, \Delta y)$ in blue are filtered from $I_j(\xi, \eta; \Delta x, \Delta y)$. The information is fully utilized to construct the object's autocorrelation $C(\Delta x, \Delta y)$ for imaging. **b** Experimental setup. A movable object is embedded in an unknown scattering medium,

which can be optically thick scattering media or thin scattering layers. A simple and speckle pattern-sized trajectory is sufficient for imaging. The front surface of the medium is illuminated via an incoherent light source, which can be either narrowband or broadband, and speckles are formed at the back surface of the medium. A series of magnified speckle images $I$ are non-invasively detected via a simple imaging system consisting of a lens and a monochrome camera. **c** Computational model. The speckle images are numerically separated into speckles and envelopes. The overlapping speckle $M_{ij}$ is the Hadamard product of any two speckles $S_i$ and $S_j$, and its sum is the overlap $C_{ij}$. The $(\Delta x_{ij}, \Delta y_{ij})$ denoted by a small arrow is calculated from the relative position between the centre and the maximum value (white point) of the cross-correlation between any two envelopes $E_i$ and $E_j$. Then, the object's autocorrelation $\Sigma C_{ij}(\Delta x_{ij}, \Delta y_{ij})$ is constructed for imaging.

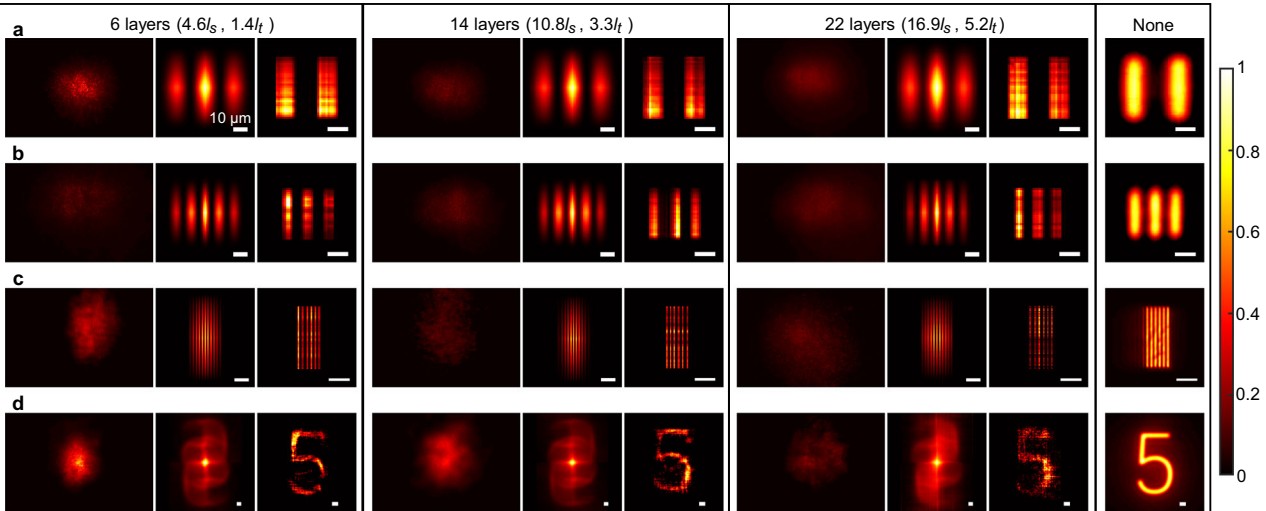

**Fig. 2 | Experimental imaging within thin to thick parafilm under single-colour LED illumination. a–d** Movable objects consisting of two 10 µm width lines (**a**), three 5 µm width lines [source data are provided as a Source data file in the Supplementary Information] (**b**), six 1 µm width lines (**c**) and a number-shaped object with a height of 100 µm (**d**) are embedded in parafilm of 6, 14, 22 and 0 layers. One of the recorded speckle images, the constructed autocorrelation and the reconstructed image are successively shown in each inset. Scale bars: 10 µm.

displacement $(\Delta x_{ij}, \Delta y_{ij})$. To further extract overlap information from the overlapping speckle, we sum it as follows:

$$
\begin{aligned}
C_{ij}(\Delta x_{ij}, \Delta y_{ij}) &= \iint M_{ij}(\xi, \eta; \Delta x_{ij}, \Delta y_{ij}) d\xi d\eta \\
&= \iint O(x,y) O(x - \Delta x_{ij}, y - \Delta y_{ij}) \iint [I_S(x,y) s(\xi, \eta; x, y)]^2 d\xi d\eta dx dy
\end{aligned}
\tag{4}
$$

The second integral $\iint [I_S(x,y) s(\xi, \eta; x, y)]^2 d\xi d\eta$ is the total energy of each impulse response, which is proportional to the energy of its point light source. Owing to low coherence and large area illumination, the contrast of $I_S(x, y)$ is low. Thus, we assume that the intensity distribution $I_S(x, y)$ within a small region containing the object is nearly uniform (Supplementary Note 5). On this basis, although the impulse responses usually have different distributions, their total energy is almost equal. Therefore, the second integral may be considered an approximately equal constant. Then, the sum of the overlapping speckle is as follows:

$$
\begin{aligned}
C_{ij}(\Delta x_{ij}, \Delta y_{ij}) &\approx \mathrm{const} \iint O(x,y) O(x - \Delta x_{ij}, y - \Delta y_{ij}) dx dy \\
&\propto O \otimes O(\Delta x_{ij}, \Delta y_{ij})
\end{aligned}
\tag{5}
$$

where $\otimes$ is the correlation operation and const denotes the constant.

According to Eq. (5), two arbitrary speckles contribute a value $C_{ij}$ to the object's autocorrelation, and its location depends on the object's relative displacement $(\Delta x_{ij}, \Delta y_{ij})$. To determine $(\Delta x_{ij}, \Delta y_{ij})$, we use the envelopes (Fig. 1 and Supplementary Fig. 12). According to the formation mechanism, the envelope moves as a whole along with the object, reflecting the object's relative displacement. To precisely extract the relative position between the slowly varying envelopes, we calculate the cross-correlation between $E_i(x, y)$ and $E_j(x-\Delta x_{ij}, y-\Delta y_{ij})$ based on the fact that they have similar but shifted distributions in terms of statistics. According to the definition of cross-correlation, $(\Delta x_{ij}, \Delta y_{ij})$ can be determined by finding the position of the maximum value relative to the centre of the cross-correlation. Combining the extracted information, a pixel of the object's autocorrelation $C_{ij}(\Delta x_{ij}, \Delta y_{ij})$ is constructed ('Methods').

**Autocorrelation construction**
To fully construct the object's autocorrelation $\Sigma C_{ij}(\Delta x_{ij}, \Delta y_{ij})$, every pixel should be calculated. By definition, a standard solution is to record all the speckle images formed by an object moving across a whole plane four times the object's area; this solution is not realistic in practice. Fortunately, the stability of incoherent speckles is high because the low spatial and temporal coherence limits the influence range of a medium's local changes on the speckles. On this basis, a simplified trajectory can also construct the object's autocorrelation as long as this trajectory contains sufficient relative positions $(\Delta x_{ij}, \Delta y_{ij})$ to sample the entire autocorrelation of the object. The sampling area is determined by the positive-valued area on the autocorrelation of the trajectory because the value of each position on the autocorrelation reflects the number of the corresponding relative position $(\Delta x_{ij}, \Delta y_{ij})$ contained in this trajectory. For example, a U or T-shaped motion trajectory with side lengths slightly larger than speckle pattern can be used for the construction of an unknown object's autocorrelation (see 'Methods' for more information).

Other trajectories, such as a rectangle, a circle or any other shapes, also work as long as its autocorrelation can cover the object's autocorrelation. In addition, a trajectory containing an L shape or even a straight line is sufficient for autocorrelation construction of axisymmetric or centrosymmetric objects, respectively. Anyway, the length of each side of the trajectory must be equal to or longer than the length of the object in the corresponding direction (Fig. 1b). For unknown objects, each side length of the trajectory should be slightly larger than the overall size of the speckle pattern, which is larger than the object size due to scattering. In addition, the detection time sequence of each position on a trajectory can be unordered. Therefore, we can record as many speckle images as possible at the shortest possible shooting intervals during the motion of an unknown and uncontrolled object, and then select a portion of the recovered trajectory for imaging. After autocorrelation construction, the object image is reconstructed by a phase-retrieval algorithm[28].

**Imaging under narrowband illumination**
As the first experimental demonstration, a single-colour light-emitting diode (LED) with 17 nm bandwidth was used as the light source (Fig. 1b). In the environment in which the isolator of the optical table was closed and the ventilation system was turned on, we selected elastic parafilm as the scattering sample to demonstrate our approach's low requirements for environment stability. Three sample thicknesses from thin to thick were used in turn; they were 6, 14 and 22 layers of parafilm corresponding to $4.6l_s$, $10.7l_s$ and $16.9l_s$, or $1.4l_t$, $3.3l_t$

and $5.2l_t$ (Supplementary Note 10). The transmissive objects were lines with widths of 10 μm, 5 μm and 1 μm and a number 5 with a height of 100 μm on negative resolution test targets (Fig. 2). Each object was moved in the middle of the parafilm sample following a U-shaped trajectory with a displacement of 1 μm between contiguous detections (except the 1 μm width object). The total detection numbers were 105, 90, 310 and 270, which were reduced by at least an order of magnitude compared with the whole-plane detections ('Methods'). The significantly reduced detection number caused speckle kinetography to have more potential use in practical applications.

One of the recorded speckle images, the constructed autocorrelation and the reconstructed image are successively shown in each inset of Fig. 2. For comparison, the objects were directly imaged by removing the parafilm. Combining the recovered images with the extracted trajectories (Supplementary Fig. 4), videos of the hidden objects were reconstructed (Supplementary Movies 1 and 2). The difference of the reconstructed images is not evident, which verifies our approach's strong ability for imaging in scattering media within thickness of $6l_t$.

In particular, we added a microscope objective before the lens because the grain size of the speckles formed by the 1 μm width lines exceeded the resolution limit of the imaging system. The object was moved 0.2 μm between contiguous detections, resulting in 5 pixels per line width in the recovered images. The experimental results verified that the imaging resolution was at least 1 μm and resists scattering of various degrees.

## Imaging under broadband illumination

In this experiment, we used a white light source (2240 nm bandwidth). Notably, the results of broadband imaging in scattering media did not suffer from broadening problems (Fig. 3), showing nearly the same results as narrowband imaging (Fig. 2). In contrast, although the same achromatic lens as the scattering imaging was used, the direct imaging results in Fig. 3 were strongly broadened due to dispersion of the white light (Supplementary Note 12).

The problem of spectral broadening during imaging is avoided because the requirement for space invariance is thoroughly released in speckle kinetography. Specifically, the impulse response formed by a white light point source is the incoherent superposition of different impulse responses at various wavelengths[29], leading to some overlaps

of impulse responses from different point sources that cannot be distinguished via a monochrome camera. In this condition, the masked speckle retains some impulse responses from point sources outside the object's overlapping region. Thus, the constructed autocorrelation becomes the following:

$$
\begin{aligned}
C_{ij}(\Delta x_{ij}, \Delta y_{ij}) &= \iint M_{ij}(\xi, \eta; \Delta x_{ij}, \Delta y_{ij})d\xi d\eta \\
&= \text{const}\,[O \otimes O(\Delta x_{ij}, \Delta y_{ij})] + B_{ij}(\Delta x_{ij}, \Delta y_{ij})
\end{aligned}
\tag{6}
$$

where $B_{ij}(\Delta x_{ij}, \Delta y_{ij})$ represents an additional fluctuation value (see 'Methods' for detailed formula derivation). Compared with the object's autocorrelation, the pixel size of $\Sigma B_{ij}(\Delta x_{ij}, \Delta y_{ij})$ is the same, but the overall size is slightly larger. Since the values at the edge of $\Sigma B_{ij}(\Delta x_{ij}, \Delta y_{ij})$ are usually small (see 'Methods', those values beyond the autocorrelation range have limited effect on the overall size of the object's autocorrelation. As a result, broadband illumination only causes a small fluctuation value error to the object's autocorrelation rather than a broadening problem. In addition, owing to the robustness of the phase-retrieval algorithm[28], the small fluctuation has little effect on the final imaging results (Fig. 3).

## Fluorescence imaging

Imaging of fluorescent beads through a chicken breast was experimentally implemented (Fig. 4a). The beads had diameters of 10 μm that were close to the size of common haemocytes. These beads were attached on the surface of a glass slide that was moved behind a chicken breast (700 μm thickness, ~3.6$l_s$, see 'Methods'). The motion trajectory was a U shape with an interval of 2 μm. The emitted fluorescence was scattered by the chicken breast, and 108 speckle images were recorded ('Methods'). The overall distribution of the reconstructed beads (Fig. 4c) was consistent with the direct imaging result (Fig. 4d), but the detail was not as good due to fewer and larger pixels of the constructed autocorrelation (Fig. 4b). As long as the speckle intensity was strong enough to be detected, fluorescent imaging of smaller beads through thicker chicken breasts could be achieved.

## Discussion

As the thickness of scattering sample increases, the speckle contrast decreases. Meanwhile, the envelope gradually broadens so that the relative displacement of the envelope will not be accurately extracted

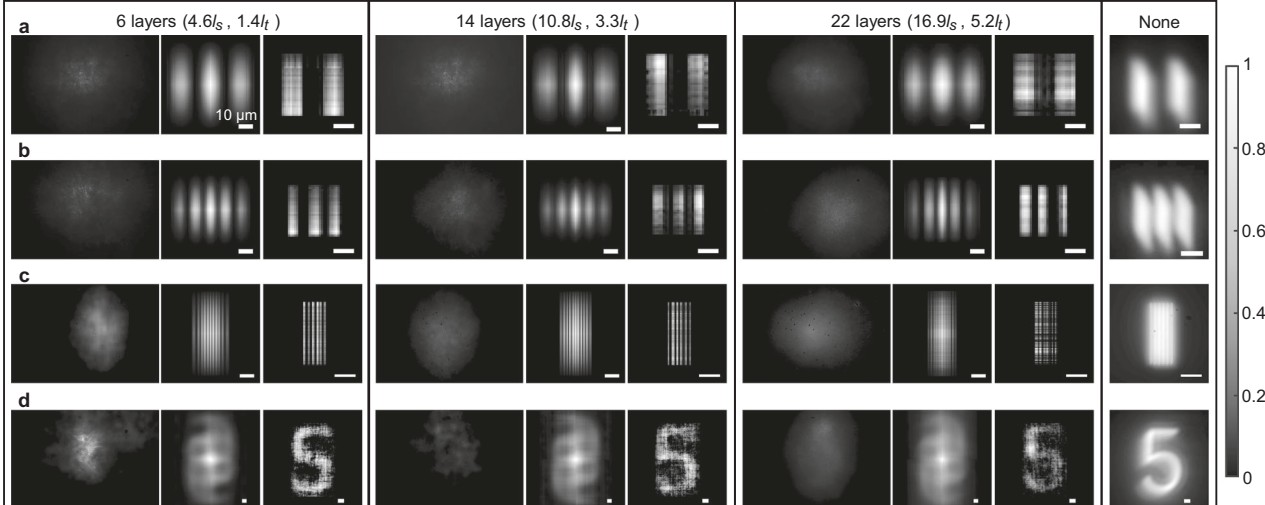

**Fig. 3 | Experimental imaging within thin to thick parafilm under white light illumination. a–d** Movable objects consisting of two 10 μm width lines (**a**), three 5 μm width lines (**b**), six 1 μm width lines (**c**) and a number-shaped object with a height of 100 μm (**d**) are embedded in parafilm of 6, 14, 22 and 0 layers. One of the recorded speckle images, the constructed autocorrelation and the

reconstructed image are successively shown in each inset. The imaging broadening problem is avoided because the space invariance is released in speckle kinetography. The imaging results without scattering medium in the path suffer from a broadening problem due to dispersion of white light. Scale bars: 10 μm.

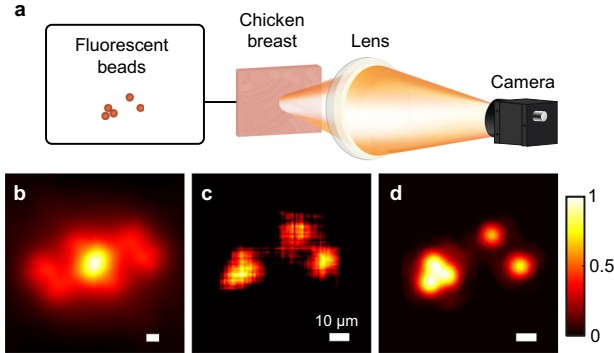

**Fig. 4 | Imaging of fluorescent beads through a chicken breast. a** Experimental setup. Fluorescence emitted from five 10 μm diameter beads is scattered by a chicken breast with a thickness of 700 μm, and the formed speckles are recorded via the imaging system consisting of a lens and a monochrome camera. **b** Constructed autocorrelation of the fluorescent beads. **c** Reconstructed image of the beads. **d** Direct imaging without chicken breast in the optical path. Scale bars: 10 μm.

especially when the entire envelope cannot be measured. These factors limit the imaging depth. Nevertheless, when the speckle contrast becomes too low to realize imaging, the entire envelope can still be measured and the relative displacement is still accurate enough for achieving the correct autocorrelation construction (see Supplementary Note 6 for experimental demonstrations). Therefore, the speckle contrast is a critical factor for the imaging depth. Under narrowband and broadband illumination, the experimental results of imaging in anisotropic and isotropic scattering samples, which respectively are parafilm and polyethylene foams[24], show that the maximum imaging depth of speckle kinetography is about $6l_t$ (Supplementary Note 6).

The following conditions are required for sharp imaging of objects via speckle kinetography. Firstly, the field of view (FOV) of the imaging system can cover the entire speckles patterns in the trajectory. Secondly, the resolution of the imaging system should be smaller than the speckle grain size. Thirdly, the interval distance of the moving object should be smaller than the narrowest line width of the object. Under the above conditions, the imaging resolution $R_{SK}$ of speckle kinetography is determined as follows:

$$R_{SK} = \begin{cases} R_{obj}(R_{IS}, L) \text{ for } R_{obj} > T_{obj} \\ T_{obj} \text{ for } R_{obj} \leq T_{obj} \end{cases} \qquad (7)$$

where $T_{obj}$ represents the interval distance of the moving object and $R_{obj}$ represents the resolution limit on the object plane. The $R_{obj}$ is determined by the resolution limit of the imaging system $R_{IS}$ and the sample thickness $L$ (Supplementary Note 7). According to the above equation, the spatial resolution can be improved by shortening the interval distance between adjacent positions on the trajectory using a high-speed camera. Furthermore, higher resolution is achievable when the imaging conditions are upgraded, including using an imaging system with higher resolution, a light source with higher brightness and a camera with higher sensitivity. The resolution limit of speckle kinetography depends on the resolution limit $R_{obj}$ on the object plane and the minimum interval distance of the object. We assume that the resolution limit $R_{IS}$ of the imaging system reaches the optical diffraction limit. Since $R_{obj}$ becomes larger as $L$ increases, $R_{obj}$ can only be close to rather than equal to the diffraction limit $R_{IS}$. Besides, the minimum interval distance of the object must be longer than the coherence length. As the interval distances in the experiments for 1-μm widths objects were already shorter than the optical diffraction limit, the minimum interval distance is not the main limit compared with

$R_{obj}$. In conclusion, the resolution limit of speckle kinetography is theoretically close to and slightly larger than the optical diffraction limit.

Although speckle kinetography has released many restrictions, it still has limitations in practice. Firstly, the medium within the region of the imaging system's FOV should keep static during the detection. The dynamic medium will cause the impulse responses' change so that the overlap information cannot be extracted from speckles anymore. Secondly, the illumination and medium within the region of the imaging system's FOV should be uniform. In this condition, the intensity distribution $I_S(x, y)$ within this region is nearly uniform (Supplementary Note 5). Otherwise, the intensity distribution of the constructed autocorrelation will be affected, which can be inferred from Eqs. (4) and (5). Thirdly, the object must be only translated but not deformed during the detection. Otherwise, the overlap information no longer represents the autocorrelation value so that the constructed object's autocorrelation will be meaningless. Fourthly, the object should be sparse. The speckle contrast decreases as the object's complexity increases since the speckle consists of more impulse responses in this case. As mentioned above, image cannot be reconstructed when the speckle contrast is very low. Thus, there is a limitation on the object's sparsity. Lastly, the contrast of the object to background should be high enough. If it is very low, the speckle contrast and the envelope contrast to background will be too low to achieve imaging. The type of the contrast can be intensity contrast, wavelength contrast and so on.

The relative displacements recovered from the envelopes are not so accurate as the existing methods[30–32]. However, a prerequisite of these methods is that the cross-correlation of objects before and after the movements must be obtained from the correlation of speckles. Therefore, the memory effect is an essential condition of these methods. As a result, the application of these methods is limited in some scenarios that the angular memory effect range is very small, such as the scattering medium is optically thick or the object is very close to the medium. In these cases, speckle kinetography can cope with these scenarios as an alternative.

Stationary objects can also be imaged through moving scattering media (Supplementary Note 8). In addition, speckle kinetography can offer a promising way to exploit other imaging techniques, e.g., non-line-of-sight (NLOS) imaging[33,34], as demonstrated in our experiment (Supplementary Note 9).

We have shown that speckle kinetography allows imaging in scattering media by taking full advantage of the movable object's information that inherently remains in the incoherent speckles. The imaging system is simple and contactless, and it only costs approximately 750 dollars ('Methods'). This general approach has the potential to shift from the laboratory to practical applications, such as imaging within or through tissue and other obscuring environments under the illumination of fluorescence, natural light or other electromagnetic wavebands.

## Methods
### Experimental setup
The experimental setup and its photograph are shown in Fig. 1b and Supplementary Fig. 1, respectively. The imaging system consists of an incoherent light source, an achromatic lens (75 mm focal length, 50.8 mm diameter, anti-reflective coating at 400-750 nm, focal length specification wavelengths at 486.1 nm, 587.6 nm and 656.3 nm) and a monochrome camera (5488 × 3672 pixels, 2.4 μm pixel size, 350 to 1000 nm response range, dynamic range of 8 bits). The light sources in the experiments were a single-colour LED (625 nm nominal wavelength, 17 nm bandwidth, 920 mW), a research arc lamp source (260 to 2500 nm wavelength, 150 W), and a continuous laser (532 nm, 480 mW) for

excitation of fluorescent beads (543 ± 40 nm excitation wavelength, 585 ± 40 nm emission wavelength). The scattering samples had gaps in their middle sections that were approximately equal to the thickness of the object and were held by a plate holder. The transmissive objects in the first two experiments were two lines (10 μm widths, 10 μm interval, 30 μm heights), three lines (5 μm widths, 5 μm intervals, 25 μm heights), six lines (1 μm widths, 1 μm intervals, 30 μm heights) and a number-shaped 5 (100 μm height). The object was embedded in the scattering samples and controlled via a two-axis motorized precision translation stage (100 nm minimum step size along the horizontal and vertical directions). To form a magnified speckle image on the camera, the distances between the back surface of the samples and the lens were within the range of 75 mm to 150 mm. The magnification of the imaging system was calculated from the object distance and image distance that was non-invasively measured in corresponding experiments and was used to resize the recovered object from the camera plane to the object plane. For the 1-μm width object, an objective (40×, NA 0.65 for thin media or 63×, NA 0.85 for thick media) was added between the samples and the lens. In the fluorescence imaging experiments, a bandpass filter (585 ± 20 nm) was mounted on the camera. The camera integration times in the different experiments were 5 ms (for large objects and thin media) to 4.8 s (for small objects and thick media). The cost of the imaging system, including the LED (283 dollars), the lens (62 dollars) and the camera (390 dollars), was only 735 dollars.

## Overlapping speckle correlation algorithm

The algorithm flow chart of the overlapping speckle correlation is shown in Supplementary Note 2. $I_i$ and $I_j$ are two arbitrary images from the recorded speckle images. After low-pass filtering in the Fourier domain, the slowly varying envelopes $E_i$ and $E_j$ are extracted from the speckle images $I_i$ and $I_j$. Then, the speckle images are divided by the corresponding envelopes to acquire the spatially normalized speckles[13] $S_i$ and $S_j$ (Supplementary Note 11). On the one hand, the Hadamard product is performed on speckles $S_i$ and $S_j$ to obtain the overlapping speckle, shown as ○ in Supplementary Fig. 2; all pixel values of the overlapping speckle are summed to obtain a value $C_{ij}$ of the object's autocorrelation. On the other hand, cross-correlation (⊗ in Supplementary Fig. 2) is performed on envelopes $E_i$ and $E_j$; the position of the maximum value is extracted to obtain the position $(\Delta x_{ij}, \Delta y_{ij})$ of value $C_{ij}$ on the object's autocorrelation. Now, a pixel $C_{ij}(\Delta x_{ij}, \Delta y_{ij})$ of the object's autocorrelation is constructed. Similarly, we keep the speckle image $I_j$ unchanged and replace the speckle image $I_i$ with $I_1$ to $I_N$. Then, a series of autocorrelation pixels $\sum_{i=1\cdots N} C_{ij}(\Delta x_{ij}, \Delta y_{ij})$ as well as the motion trail of the object are calculated in the same way. Finally, all pixels of the object autocorrelation $\sum_{i=1\cdots N, j=1\cdots N} C_{ij}(\Delta x_{ij}, \Delta y_{ij})$ are obtained by calculating every two of the detected speckle images. The object image $O$ is then reconstructed from the constructed autocorrelation through the Fienup-type iterative phase-retrieval algorithm[28] (Supplementary Note 13). The code of overlapping speckle correlation algorithm is provided in the Supplementary Code 1.

## Autocorrelation construction with a simple trajectory

Under incoherent illumination, the object's autocorrelation can be constructed with a simple trajectory with side lengths slightly larger than the speckle pattern. For example, speckle images generated from a moving object are recorded. According to the cross-correlations between every two envelopes filtered from the speckle images, all the relative positions of the object are extracted, which consist the recovered trajectory of the object (Supplementary Fig. 3a). The side lengths of the trajectory are slightly larger than the overall size of the speckle pattern, which is larger than the object size due to scattering. The autocorrelation of this

U-shaped trajectory is shown in Supplementary Fig. 3b. The value of each position on this autocorrelation reflects the number of the corresponding relative position $(\Delta x_{ij}, \Delta y_{ij})$ contained in this trajectory. Therefore, according to Eq. (5), the object's autocorrelation can be sampled at the positive-valued positions of the trajectory's autocorrelation. As shown in Supplementary Fig. 3c, in the positive-valued area of the trajectory's autocorrelation, the object's autocorrelation is completely sampled. The value of each pixel in the sampling area are calculated from the corresponding spatially normalized speckles according to Eq. (5). Besides, as the object's line width is not known, the interval between every two shots is set as small as possible to ensure the resolution for imaging (Supplementary Fig. 3d).

In actual detections, the object is unknown and uncontrolled. In this case, we record as many speckle images as possible at the shortest possible shooting intervals. According to the cross-correlations between every two envelopes, the trajectory is recovered (Supplementary Fig. 3e). Then, we select a T-shaped portion of the trajectory for computational processing (blue dotted T shape in Supplementary Fig. 3e) since its side lengths are slightly larger than the speckle pattern. As shown in Supplementary Figs. 3f and 3g, the autocorrelation of this T-shaped trajectory is large enough to cover and sample the entire autocorrelation of the object. Based on this, the object image is reconstructed (Supplementary Fig. 3h). Besides, since the motion interval of spontaneously moving objects cannot be fully controlled, the imaging resolution requirements may not be met in some extreme cases.

The U and T shaped trajectories are just two workable examples. In fact, no matter what shape the trajectory is, as long as its autocorrelation can cover the entire autocorrelation of the object with sufficient resolution, the object image can be reconstructed. If the computational capacity is sufficient, the whole trajectory shown in Supplementary Fig. 3e can be utilized for imaging without selection. Therefore, the only requirement is recording speckle images as many and fast as possible during the object motion, regardless of the object's shape or motion trajectory.

## Recovered trajectories

The trajectories of the moving objects in the experiments are recovered and partially shown in Supplementary Fig. 4. As the overall width and height of the 5 μm width object (Fig. 2b) are both 25 μm, the detection number of the speckle images should be more than 25 in every direction when the object displacement between contiguous detections is 1 μm. In the first two experiments, 30 speckle images from this object with 1 μm intervals were detected on each side of the U-shaped trajectory. The object's trajectory is recovered from these speckle images via cross-correlation of the envelopes (Supplementary Fig. 4b). The total detection number of the speckle images is 90. In contrast, if we record all speckle images formed by the object moving across the whole plane, the detection number would be 900 for the same resolution.

In the experiments, the detection numbers of the speckle images generated from the objects consisting of two 10 μm width lines, a number-shaped 5, five fluorescent beads and six 1 μm width lines were 105, 270, 108 and 310. In contrast, the number of whole-plane detections would be 1225, 8800, 1296 and 11,900 for the same resolutions. Our detection number is reduced by at least an order of magnitude. The low requirements for detection number and motion trajectory cause speckle kinetography to have more potential use in practical applications.

## Formula derivation for broadband imaging

Under white light illumination, the nonoverlapping assumption of impulse responses from different point sources cannot be rigorously implemented. In this condition, the overlapping speckle

$M_{ij}(\xi,\eta; \Delta x_{ij},\Delta y_{ij})$ in Eq. (3) consists of two parts as shown below:

$$
\begin{aligned}
& M_{ij}(\xi,\eta; \Delta x_{ij},\Delta y_{ij}) \\
& = S_i(\xi,\eta)S_j(\xi,\eta; \Delta x_{ij},\Delta y_{ij}) \\
& = \iint\iint O(x',y')I_S(x',y')s(\xi,\eta; x',y') \\
& \quad \times O(x-\Delta x_{ij}, x-\Delta y_{ij})I_S(x,y)s(\xi,\eta; x,y)dx'dy'dxdy \\
& = \iint O(x,y)O(x-\Delta x_{ij}, y-\Delta y_{ij})I_S^2(x,y)s^2(\xi,\eta; x,y)dxdy \\
& + \iint_{x\neq x', y\neq y'} O(x',y')O(x-\Delta x_{ij}, y-\Delta y_{ij})I_S(x',y')I_S(x,y) \\
& \quad \times s(\xi,\eta; x',y')s(\xi,\eta; x,y)dx'dy'dxdy
\end{aligned}
\tag{8}
$$

which are the product of overlapped impulse responses from the same and the different point sources, respectively. Therefore, the sum of the overlapping speckle becomes the following:

$$
\begin{aligned}
& C_{ij}(\Delta x_{ij},\Delta y_{ij}) \\
& = \iint M_{ij}(\xi,\eta; \Delta x_{ij},\Delta y_{ij})d\xi d\eta \\
& = \iint O(x,y)O(x-\Delta x_{ij}, y-\Delta y_{ij})I_S^2(x,y)\iint s^2(\xi,\eta; x,y)d\xi d\eta dxdy \\
& + \iint_{x\neq x', y\neq y'} O(x',y')O(x-\Delta x_{ij}, y-\Delta y_{ij})I_S(x',y')I_S(x,y) \\
& \quad \times \iint s(\xi,\eta; x',y')s(\xi,\eta; x,y)d\xi d\eta dx'dy'dxdy \\
& \approx \mathrm{const}\iint O(x,y)O(x-\Delta x_{ij}, y-\Delta y_{ij})dxdy + B_{ij}(\Delta x_{ij},\Delta y_{ij}) \\
& = \mathrm{const}\,[O\otimes O(\Delta x_{ij},\Delta y_{ij})] + B_{ij}(\Delta x_{ij},\Delta y_{ij})
\end{aligned}
\tag{9}
$$

where and represents an additional intensity fluctuation of the constructed object's autocorrelation, which is caused by the overlap of the impulse responses from different point sources and does not contain overlap information. Then, we obtain the derivation result of Eq. (6).

Indeed, $B_{ij}(\Delta x_{ij},\Delta y_{ij})$ occurs not only when the illumination is broadband, but also when the scattering medium is optically thick. The root cause is the decrease of speckle contrast in these conditions. $B_{ij}(\Delta x_{ij},\Delta y_{ij})$ is usually a bell shape, like the autocorrelation of a halo. Therefore, the autocorrelation constructed from low-contrast speckles often likes an object's autocorrelation fluctuating on the surface of a bell-shaped distribution. And the slope of the bell-shaped distribution increases with the decrease of speckle contrast. Thus, the contrast of the constructed autocorrelation decreases with the speckle contrast. Nevertheless, the object images recovered through the phase-retrieval algorithm[13,28] are little affected, because the low-contrast speckle only slightly affects the contrast rather than the structure of the object's autocorrelation, which have limited impact on the phase-retrieval algorithm with constraints of realness and non-negativity (Supplementary Note 13 and Supplementary Box 1).

## Data availability
The minimum dataset required to reproduce the findings of this study can be found in this article and its Supplementary Information files, and remaining data are available from the authors. Source data are provided with this paper.

## Code availability
Code related to the proposed method can be downloaded from the Supplementary Information.

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

## Acknowledgements

We acknowledge useful discussions with Meiling Zhang, Ming She and Kai Gao. We also acknowledge Jiaqing Liu for her helpful suggestions in composition and rendering of Fig. 1. This work was supported by the National Natural Science Foundation of China (62105146 Y.S., 12274225 Y.F., 12274224 Y.L.), the Natural Science Foundation of Jiangsu Province (BK20210290 Y.S., BK20230089 Y.F.) and the Fundamental Research Funds for Central Universities (NS2022079 Y.S.).

## Author contributions

Y.S. conceived the idea, wrote the overlapping speckle correlation algorithm and designed the experiments. All the authors conducted the theoretical analysis. Y.S. and W.S. performed the experiments and analysed the results. Y.S. and Y.F. wrote the manuscript with inputs from all authors. Y.S., Y.F. and Y.L. supervised the project.

## Competing interests

The authors declare no competing interests.
