## [Peer Review File · Nature Communications]

Overlapping speckle correlation algorithm for high-resolution imaging and tracking of objects in unknown scattering mediaREVIEWER COMMENTS

Reviewer #1 (Remarks to the Author):

The authors proposed a non-invasive imaging method for a hidden moving object under incoherent illumination. They measured incoherent speckle images at various object positions. They utilized low-pass filtered speckle images and the "speckle masking" operation to find the auto-correlation of the hidden object. Then, a phase retrieval algorithm is used to retrieve the image of the object from the auto-correlation.

The method is an extension of previous studies (Particularly, Ref. 12, 17 and [Nature communications, 13(1), 5779.]). Its key differences are the use of the broadband incoherent illumination and object's motion trajectory. However, because these benefits are only possible under specific conditions, I am not sure if the proposed method truly addresses the limitations as the manuscript claims. Furthermore, there are several points that remain unclear regarding the underlying assumptions and data processing, and the manuscript does not discuss the limitations stemming from these assumptions. Below are my remarks.

- My main concern is with the claims related to the motion trajectory. The authors claimed that the trajectory can be spontaneous, and simple U- or T-shaped trajectories are sufficient for this method. However, the trajectory determines which area of the auto-correlation can be measured (Eq. 5). In fact, the auto-correlation can only be sampled at the position described by the auto-correlation of the trajectory. Thus, it is impossible to discuss which trajectory (shape, size, and the number of sampling points along the trajectory) is enough to represent the auto-correlation without a prior knowledge about the object. The authors' demonstration on U- and T-shaped trajectories (Extended Figs. 3-4) are based on a simple and sparse shape of the auto-correlation. These points contradict the claim made in lines 52-54 that previous techniques require prior knowledge of the hidden objects and a trajectory, and the proposed method addresses the issues.

- The underlying assumption of this method is that the entire speckle should be measured. This is essential as the envelope plays an important role in determining the displacement of an object. However, with a highly scattering medium, measuring the entire speckle is not an

easy task. For example, in a case where speckle images extend over a large area, the proposed method will not work. In this regard, the authors' claim about "having unlimited imaging depth" is an overstatement. Furthermore, the experiments were conducted with (more or less) forward scattering media, such that the entire speckle could be measured. While I agree parafilm layers have "thickness of 4.6 to 16.9 scattering mean free paths", the abstract can be misleading because any medium with similar thickness but with low anisotropic factors can make it impossible to measure the envelope properly.

- In Eq. (3), the authors assumed that "the impulse responses do not overlap with each other". However, because $s(x,y)$ is low-contrast speckles with a positive mean value, the orthogonality between $s(x',y')$ and $s(x,y)$ in Eq. (8) does not hold. This could make the autocorrelation inaccurate, making phase retrieval unstable. Considering that phase retrieval is highly sensitive to the auto-correlation, it is unclear how this noise term can be easily neglected without additional processing.

- Related to my previous comment, the normalized speckle S is obtained by dividing the speckle image with its envelope. The odd thing is that the values outside the envelope has near 0 value. This will make S have noisy values outside the envelope. However, in Extended Fig.2, S also shows near 0 value outside the envelope. This part needs more explanation.

- There are other techniques to image/track hidden objects such as [Applied optics, 57(4), 905-913., IEEE Photonics Journal, 11(6), 1-14., Nature communications, 13(1), 5779.]. The authors should discuss the difference between their work and the existing studies.

- The term "speckle masking" had already been used for the bispectrum analysis of speckles [Applied Optics, 22(24), 4028-4037.]. Using the term for this work can be confusing since there is non-invasive imaging of moving objects based on the bispectrum analysis [IEEE Photonics Journal, 11(6), 1-14.].

- For the broadband experiment, the authors claimed that the proposed method can avoid "image broadening problem". But it is unclear why an object imaged without a scattering medium is blurry, as in Fig. 3(d). The broadening appears to be due to the chromatic

aberration of optics, which can easily be avoided with proper lenses. Furthermore, in lines 253-255, the authors described that the proposed method works without a medium. However, in such a case, the direct image of an object can be easily obtained. I am not sure if these are important points to discuss.

- It is not clear which phase retrieval algorithm was used. Fienup-type phase retrieval can be any algorithm that employs an object plane constraint. The authors need to explain this in more detail.

- In line 216, it is unclear what “edge value” and “the excess edge can be disregarded” means.

- The authors should include the NAs of objective lenses used.

- Regarding Eq. (5), I suggest the authors to rewrite the autocorrelation to $O^*O(\Delta x)$ instead of $O(\Delta x)^*O(\Delta x)$.

Reviewer #2 (Remarks to the Author):

In this manuscript the authors propose a new approach in the family of speckle autocorrelation imaging techniques to visualize objects that are hidden within a scattering medium. Originally proposed by Katz et al. in Nature Photonics (2014) and Bertolotti et al. in Nature (2012), this category of techniques utilizes speckle patterns to reconstruct an object's 2D autocorrelation. Images are then recovered using a phase-retrieval algorithm.

In the present work, authors demonstrate the utility of this approach for objects that are moving inside the scattering medium and that surprisingly few recordings are needed to sample the objects' autocorrelation. To acquire the raw data, they move the sample along at least two axes (e.g. a T-shaped or U-shaped pattern) and record speckle pattern for each location. To recover the object's 2D autocorrelation, they use low-pass filtering to estimate the relative shift between two recordings and they calculate the dot product between the high-pass filtered speckle patterns to get the value of the autocorrelation at this shift. Once

an object's 2D autocorrelation is obtained, they use phase-retrieval to recover the image.

I find this work to be conceptually interesting and relevant to the field of imaging in scattering media. The description of the experiments and data analysis is clear and should make the paper easy to follow and reproduce. However, the authors should address the following issues before I can support publication:

Major comments:

Starting with the abstract and in multiple places in the manuscript, the authors claim that their technique allows imaging inside scattering media of arbitrary (unlimited) thickness with diffraction-limited resolution. However, this method critically depends on two factors that degrade with medium thickness: first, an accurate estimation of the relative shift of the envelopes and second, speckle contrast. These limitations should be discussed in detail and the manuscript's claims should be adjusted accordingly.

Related: How does sample thickness affect the precision with which the shifts of the object autocorrelation function are reconstructed? Given that the authors claim that the tissue thickness did not visibly affect the object reconstruct, could it be attributed to the scattering anisotropy of the samples (parafilm)? In this case, the transport mean free path of the sample would be much greater than the scattering mean free path, and thus the memory effect will be present at much greater depths. Would this method still work when using a strong diffuser? In a similar fashion, strongly diffusive samples would exhibit a much stronger spectral broadening, blurring the resulting speckle and making the reconstruction for broadband illumination impossible.

Minor comments:

Please include the example recorded intensity images in Figs. 1 and 2 next to the reconstructed autocorrelations.

Please remove the sentence "Interestingly, speckle kinetography works even when there is

no medium in the optical path because the impulse response degenerates to a point. In a sense, the effect of the scattering medium on imaging is eliminated." from the Discussion as this is self-evident.

Reconstructed trajectories in Extended Fig. 5 are hardly visible. Could the authors plot them more visibly and add the real trajectories for comparison?

The choice of the name "speckle masking" isn't obvious to me, when the operation is essentially a dot product of two high-pass filtered speckle patterns ("On the one hand, the Hadamard product is performed on speckles S_i and S_j to implement the speckle-masking operation, shown as 'o' in Extended Data Fig. 2; all pixel values of the masked speckle are summed to obtain a value C_{ij} of the object's autocorrelation")

The practical usefulness of this method seems to be limited by (a) the assumption of uniform object intensity, (b) the requirement for object sparsity, (c) the requirement of object stability (no transformation other than translation). In my opinion this paper is still an important and interesting contribution without solving these problems, but the authors should give the reader a detailed explanation of these limitations and illustrate where the method starts to fail.

Related: some sections imply a wide range of possible applications ("The motion of the object can be either spontaneous, such as blood cells under tissue, aircraft upon clouds and creatures under ice cover, or controlled, such as acoustic manipulation for cells, optical forces for particles and remote operation for drones.") which could mislead the reader as none of these examples are currently within reach of this method. Authors should either remove such examples or state explicitly that these examples are beyond the limits of the method (or provide experimental evidence to the contrary).

Reviewer #3 (Remarks to the Author):

Shi et al. presented a technique termed Speckle kinetography for imaging a moving object within scattering media. The main concept involves recording the pattern of light

transmitted through the scattering medium while the object moves. Information about the object's autocorrelation is obtained from the correlation between these patterns. This information is then used to retrieve the object image through a phase retrieval algorithm. The technique was demonstrated at a proof-of-concept level with samples such as vertical bars, number patterns, and a few fluorescent particles. Scattering media such as parafilm or chicken breast tissue were used, and the thickness of the scattering medium ranged up to 16.9 times the scattering mean free path. The resolution of the reconstructed object was about 1 micrometer.

While the idea itself is new, it only works under very restricted conditions. The entire object within the scattering medium must move in a specific pattern, which is rare in practice. Furthermore, its trajectory must be precisely known to construct the object's autocorrelation map, which is said to be inferred from the background envelope. However, this is only possible when a bright object is spatially isolated. Additionally, obtaining object information from its autocorrelation map mostly applies to objects with structures as simple as those demonstrated in the study with high contrast. Typically, autocorrelation tends to reduce the contrast of an object's fine structures as spatial information about the object gets integrated, making it highly sensitive to noise. In this context, the proposed method of obtaining an object's autocorrelation from the correlation of its incoherent images appears applicable mainly to objects with high contrast. Considering these factors, the study lacks a broad impact and potential for applicability. Therefore, it would be suitable for a more specialized journal.

Here are a few comments that need consideration:

1. The layout of Fig. 1 seems ambiguous to properly convey the concept. It would be better to arrange the subpanels in a way that directly represents Equations 1 and 2.
2. In actual measurements, what is the ratio of the envelope to the object's autocorrelation component? And what determines this?
3. The position of the object is inferred from the envelope, a non-interfering component. This method ultimately relies on detecting the displacement of the envelope due to the object's movement. This process seems to require an object to be isolated. Additionally, if

the envelope is too broad, the precision of detecting movement could decrease.

4. The resolution of image reconstruction appears to be determined by the step of the object's movement, and the resolution and field of view of the imaging system. A theoretical model and analysis are needed.

5. Parafilm and chicken breast tissue were used as the scattering media, but what is the distance between the object and the scattering medium? In fact, if the distance is large, high-angle multiple scattering can effectively be filtered out, which leads to an overestimation of the effect of the scattering medium. Typically, objects are not separated from the scattering medium, but embedded within it.

Response to reviewers

We appreciate it very much for all the valuable comments from the reviewers. These comments are helpful for improving the paper and have important guiding significance to our researches. In this letter, we provide specific responses to these comments and list the corresponding modifications made to the manuscript.

Reviewer 1:

1. Reviewer comment

The authors proposed a non-invasive imaging method for a hidden moving object under incoherent illumination. They measured incoherent speckle images at various object positions. They utilized low-pass filtered speckle images and the "speckle masking" operation to find the auto-correlation of the hidden object. Then, a phase retrieval algorithm is used to retrieve the image of the object from the auto-correlation.

The method is an extension of previous studies (Particularly, Ref. 12, 17 and [Nature communications, 13(1), 5779.]). Its key differences are the use of the broadband incoherent illumination and object's motion trajectory. However, because these benefits are only possible under specific conditions, I am not sure if the proposed method truly addresses the limitations as the manuscript claims. Furthermore, there are several points that remain unclear regarding the underlying assumptions and data processing, and the manuscript does not discuss the limitations stemming from these assumptions. Below are my remarks.

My main concern is with the claims related to the motion trajectory. The authors claimed that the trajectory can be spontaneous, and simple U- or T-shaped trajectories are sufficient for this method. However, the trajectory determines which area of the auto-correlation can be measured (Eq. 5). In fact, the auto-correlation can only be sampled at the position described by the auto-correlation of the trajectory. Thus, it is impossible to discuss which trajectory (shape, size, and the number of sampling points along the trajectory) is enough to represent the auto-correlation without a prior knowledge about the object. The authors' demonstration on U- and T-shaped trajectories (Extended Figs. 3-4) are based on a simple and sparse shape of the auto-correlation. These points contradict the claim made in lines 52-54 that previous techniques require prior knowledge of the hidden objects and a trajectory, and the proposed method addresses the issues.

Response

We thank the reviewer for the thorough review and constructive comments. We agree that “the autocorrelation can only be sampled at the position described by the autocorrelation of the trajectory” is a very accurate and concise description. In light of this description, we would like to reinterpret the autocorrelation construction based on a simplified trajectory.

Fig. R1 Autocorrelation construction based on a simple trajectory. The object image can be reconstructed as long as the autocorrelation of the trajectory is large enough to cover the object’s autocorrelation with sufficient resolution. **a**, Direct imaging of an object moving in a U shape. The trajectory and the object are of the same length and width. **b** and **c**, Autocorrelation of the trajectory in (**a**) shown with different colour bars. **d**, The constructed autocorrelation of the object. The sampling area is determined by the autocorrelation of the trajectory in (**c**). **e**, The recovered object image. **f**, Speckle images from a hidden object with a U-shaped trajectory. The side lengths of the trajectory are slightly larger than the speckle pattern. **g** and **h**, Autocorrelation of the trajectory in (**f**) shown with different colour bars. **i**, The constructed autocorrelation of the object with a sampling area determined by (**h**). **j**, The recovered object image. **k**, Speckle image and trajectory from a spontaneously moving object. **l**, The partially selected trajectories in a T shape. **m**, Autocorrelation of the trajectory in (**l**). **n**, The constructed autocorrelation of the object with a sampling area determined by (**m**). **o**, The object image recovered from (**n**). **p**, Speckle images from a spontaneously moving object. **q**, The partially selected trajectories in a small U shape. **r**, Autocorrelation of the trajectory in (**q**). **s**, The constructed autocorrelation of the object with a sampling area determined by (**r**). **t**, The unsuccessfully recovered object image. Scale bars: 50 μm .

As an example, an object consisting of five letters is moved in a U-shaped trajectory (Fig. R1a). The trajectory and the object are of the same length and width. The autocorrelation of the trajectory is shown in Fig. R1b. The value of each position on the autocorrelation reflects the number of the corresponding relative position (Δx_{ij} , Δy_{ij}) contained in this trajectory. Therefore, according to Eq. (5), the object's autocorrelation can be sampled at the positive-valued positions of the trajectory's autocorrelation. To visualize all the positive-valued positions, the colour bar range of the trajectory's autocorrelation is adjusted (Fig. R1c). As shown in Fig. R1d, the object's autocorrelation is completely sampled within the positive-valued area of the trajectory's autocorrelation in Fig. R1c. Then, the object image can be reconstructed through a phase retrieval algorithm (Fig. R1e).

When the object is hidden inside a scattering medium, we cannot know the size of the object. Although we have no prior knowledge about the object, we can choose a U-shaped trajectory with length and width slightly larger than the overall size of the speckle pattern (Fig. R1f). The overall size of the speckle pattern is larger than the object size due to scattering. Therefore, this trajectory ensures that the sampling area (Figs. R1g and R1h) is large enough to cover the object's autocorrelation (Fig. R1i). Besides, since the object's line width is not known, the interval distance between every two shots is set as small as possible to ensure the resolution for imaging (Fig. R1j). And the number of sampling points is calculated by dividing the trajectory length by the interval distance. Since the smallest step length of our system is smaller than the resolution limit of our imaging system, it is sufficient to support the reconstruction of objects that our system can image.

In actual detections, the object is unknown and uncontrolled. In this case, we need to record as many speckle images as possible at the shortest possible shooting intervals during the object motion. According to the positions extracted from the cross-correlations between every two envelopes filtered from the speckle images, a trajectory is recovered (Fig. R1k). Then, a portion of the trajectory is selected for computational processing. Based on the selection principle mentioned above, we choose trajectories 4 and 6 (Fig. R1l) since their side lengths are slightly larger than the speckle pattern. As shown in Figs. R1m and R1n, the autocorrelation of this T-shaped trajectory is large enough to sample the whole object's autocorrelation for imaging (Fig. R1o). As a contrast, if we choose trajectories 1 to 3 (Fig. R1q), the autocorrelation of this trajectory (Fig. R1r) cannot cover the whole object's autocorrelation (Fig. R1s), leading to the failure of object reconstruction (Fig. R1t). Besides, since the object motion is uncontrolled, the motion interval depends on the displacement between shots. In some extreme cases, even if we shoot continuously as fast as possible, the step length may

still not meet the imaging resolution requirements, which would lead to blurred imaging.

Notably, the trajectories in U and T shapes are just two workable examples. In fact, no matter what shape the trajectory is, as long as its autocorrelation can cover an object's autocorrelation with sufficient resolution, the object image can be reconstructed. If the computational capacity is sufficient, the whole trajectory in Fig. R1k can be utilized for imaging without selection. Therefore, the only requirement is recording speckle images as many and fast as possible, regardless of the object's shape or motion trajectory.

Action taken

We have revised the "Autocorrelation construction" section in the manuscript inspired by the reviewer's brilliant description of "sampling at autocorrelation of the trajectory" as follows:

"To fully construct the object's autocorrelation $\Sigma C_{ij}(\Delta x_{ij}, \Delta y_{ij})$, every 'pixel' should be calculated. By definition, a standard solution is to record all the speckle images formed by an object moving across a whole plane four times the object's area; this solution is not realistic in practice. Fortunately, the stability of incoherent speckles is high because the low spatial and temporal coherence limits the influence range of a medium's local changes on the speckles. On this basis, a simplified trajectory can also construct the object's autocorrelation as long as this trajectory contains sufficient relative positions $(\Delta x_{ij}, \Delta y_{ij})$ to sample the entire autocorrelation of the object. The sampling area is determined by the positive-valued area on the autocorrelation of the trajectory because the value of each position on the autocorrelation reflects the number of the corresponding relative position $(\Delta x_{ij}, \Delta y_{ij})$ contained in this trajectory. For example, a 'U' or 'T' shaped motion trajectory with side lengths slightly larger than speckle pattern can be used for the construction of an unknown object's autocorrelation (see Methods for more information).

Other trajectories, such as a rectangle, a circle or any other shapes, also work as long as its autocorrelation can cover the object's autocorrelation. Additionally, a trajectory containing an 'L' shape or even a straight line is sufficient for autocorrelation construction of axisymmetric or centrosymmetric objects, respectively. Anyway, the length of each side of the trajectory must be equal to or longer than the length of the object in the corresponding direction (Fig. 1b). For unknown objects, each side length of the trajectory should be slightly larger than the overall size of the speckle pattern, which is larger than the object size due to scattering. In addition, the detection time sequence of each position on a trajectory can be unordered. Therefore, we can record as many speckle images as possible at the shortest possible shooting intervals during

the motion of an unknown and uncontrolled object, and then select a portion of the recovered trajectory for imaging. After autocorrelation construction, the object image is reconstructed by a phase-retrieval algorithm²⁸.”

Accordingly, the two sections “Autocorrelation construction with a ‘U’/ ‘T’ shaped trajectory” in Methods and the corresponding Extended Data Figs. 3 and 4 have been deleted because these descriptions were unclear and cumbersome. And we have added a section “Autocorrelation construction with a simple trajectory” in the Methods as well as the Supplementary Fig. 3 in the manuscript as follows:

“Autocorrelation construction with a simple trajectory

Under incoherent illumination, the object’s autocorrelation can be constructed with a simple trajectory with side lengths slightly larger than the speckle pattern. For example, speckle images generated from a moving object are recorded. According to the cross-correlations between every two envelopes filtered from the speckle images, all the relative positions of the object are extracted, which consist the recovered trajectory of the object (Supplementary Fig. 3a). The side lengths of the trajectory are slightly larger than the overall size of the speckle pattern, which is larger than the object size due to scattering. The autocorrelation of this U-shaped trajectory is shown in Supplementary Fig. 3b. The value of each position on this autocorrelation reflects the number of the corresponding relative position $(\Delta x_{ij}, \Delta y_{ij})$ contained in this trajectory. Therefore, according to Eq. (5), the object’s autocorrelation can be sampled at the positive-valued positions of the trajectory’s autocorrelation. As shown in Supplementary Fig. 3c, in the positive-valued area of the trajectory’s autocorrelation, the object’s autocorrelation is completely sampled. The value of each pixel in the sampling area are calculated from the corresponding spatially normalized speckles according to Eq. (5). Besides, as the object’s line width is not known, the interval between every two shots is set as small as possible to ensure the resolution for imaging (Supplementary Fig. 3d).

In actual detections, the object is unknown and uncontrolled. In this case, we record as many speckle images as possible at the shortest possible shooting intervals. According to the cross-correlations between every two envelopes, the trajectory is recovered (Supplementary Fig. 3e). Then, we select a T-shaped portion of the trajectory for computational processing (blue dotted T shape in Supplementary Fig. 3e) since its side lengths are slightly larger than the speckle pattern. As shown in Supplementary Figs. 3f and 3g, the autocorrelation of this T-shaped trajectory is large enough to cover and sample the entire autocorrelation of the object. Based on this, the object image is reconstructed (Supplementary Fig. 3h). Besides, since the motion interval of spontaneously moving objects cannot be fully controlled, the imaging resolution requirements may not be met in some extreme cases.

The U and T shaped trajectories are just two workable examples. In fact, no matter what shape the trajectory is, as long as its autocorrelation can cover the entire autocorrelation of the object with sufficient resolution, the object image can be reconstructed. If the computational capacity is sufficient, the whole trajectory shown in Supplementary Fig. 3e can be utilized for imaging without selection. Therefore, the only requirement is recording speckle images as many and fast as possible during the object motion, regardless of the object's shape or motion trajectory.

Supplementary Fig. 3 Autocorrelation construction with a simple trajectory. The object image can be reconstructed as long as the autocorrelation of the trajectory is large enough to cover the object's autocorrelation with sufficient resolution. **a**, Speckle image from a hidden object. The trajectory extracted from the cross-correlations between every two envelopes is in a U shape. The overall size of the trajectory is slightly larger than the speckle pattern. **b**, Autocorrelation of the U-shaped trajectory in (a). **c**, The constructed autocorrelation of the object with a sampling area determined by (b). **d**, The object image recovered from (c). **e**, Speckle image from a spontaneously moving object. The trajectory is recovered from the cross-correlations between every two envelopes. **f**, Autocorrelation of the partially selected trajectories shown as a blue dotted T shape in (e). **g**, The object's autocorrelation with a sampling area determined by (f). **h**, The object image recovered from (g). Scale bars: 50 μm ."

Since the imaging resolution of spontaneously moving object cannot be guaranteed in some extreme cases, we have deleted all the statements of “the motion can be spontaneous” in the manuscript. Instead, we have discussed the method for imaging of spontaneously moving objects in the Methods.

To avoid ambiguity, in the Introduction of the revised manuscript, the sentence in lines 52-58 has been revised as follows:

“Although submillimetre resolution imaging of moving objects through unknown heavily scattering media has been achieved, applying these approaches in practical scenarios is still challenging, since they introduce extra problems caused by the indispensable requirements of a motion trajectory across a whole plane, prior knowledge regarding the hidden object's positions and a specific coherent light source.”

In the Introduction of the revised manuscript, the description of “a U-shaped motion trajectory” has been revised as “a simple and speckle pattern-sized motion trajectory”.

2. Reviewer comment

The underlying assumption of this method is that the entire speckle should be measured. This is essential as the envelope plays an important role in determining the displacement of an object. However, with a highly scattering medium, measuring the entire speckle is not an easy task. For example, in a case where speckle images extend over a large area, the proposed method will not work. In this regard, the authors' claim about “having unlimited imaging depth” is an overstatement. Furthermore, the experiments were conducted with (more or less) forward scattering media, such that the entire speckle could be measured. While I agree parafilm layers have “thickness of 4.6 to 16.9 scattering mean free paths”, the abstract can be misleading because any medium with similar thickness but with low anisotropic factors can make it impossible to measure the envelope properly.

Response:

We thank the reviewer for the important reminder of imaging depth limit and would like to explore it by implementing two groups of experiments.

Fig. R2 The influence of medium thickness on three factors. **a**, An object consists of three lines with $25\ \mu\text{m}$ width and $125\ \mu\text{m}$ height. **b**, The SSIM of the constructed autocorrelation versus the thickness L of parafilm. **c**, The FWHM W of the envelopes versus L . **d**, The speckle contrast versus L . Scale bar: $50\ \mu\text{m}$.

In the first group of experiments, the scattering samples were parafilm of 2 to 30 layers, in steps of 2 layers. As shown in Fig. R2a, a transmissive object consisting of three lines with $25\ \mu\text{m}$ width and $125\ \mu\text{m}$ height was embedded in the middle of the parafilm. This object was larger than the objects described in the manuscript so that sufficient light intensity could be detected even when the parafilm was thicker than 22 layers. In each sample thickness, 52 speckle images from this axisymmetric object moved in an L shape were recorded, in steps of $5\ \mu\text{m}$. According to the recorded 780 speckle images, three curves were calculated (Figs. R2b-R2d).

The first curve (Fig. R2b) shows that the structural similarity index measurement (SSIM) of the constructed autocorrelation decreases as the sample thickness L increases. The downward trend of the SSIM is relatively slow from 2 to 26 layers, and turns steep from 26 to 30 layers. When the sample thickness reaches 28 layers, the object's

autocorrelation becomes too blurry to recover the object image. Therefore, the maximum imaging depth here is about 26 layers of parafilm, corresponding to 19.9 scattering mean free path l_s and 6.2 transport mean free path l_t , which is calculated from the experimentally measured anisotropy coefficient g of 0.69.

Next, we would like to determine the main limiting factors according to the other two curves. The curve in Fig. R2c shows that the full width at half-maximum (FWHM) of the envelope W broadens as the sample thickness L increases. But the entire envelope can still be measured when the parafilm is 28 layers. In addition, the magnification of the imaging system can be adjusted within a certain range to measure the entire envelopes in most experiments. Thus, the expansion of the envelope is not the main limitation here. The curve in Fig. R2d shows that the speckle contrast decreases as the sample thickness L increases. Moreover, the speckle contrast quickly drops below 0.1 from 26 to 30 layers. Owing to the extremely low contrast, the speckles overlap with each other even when the corresponding objects do not overlap. It causes the small values in the object's autocorrelation to become large, which is consistent with the blurred autocorrelation constructed at 28 layers (Fig. R2b). In conclusion, both the envelope expansion and the speckle contrast decrease limit the imaging depth, and the speckle contrast decrease dominates.

Fig. R3 Imaging of an object embedded in 15 mm thick polyethylene foams. **a** and **b**, Front (**a**) and side (**b**) photographs of the polyethylene foams. **c**, A recorded speckle image. **d**, The constructed autocorrelation of the object. **e**, The reconstructed object image. Scale bars: 50 μm .

In the second group of experiments, the scattering samples were polyethylene foams (Figs. R3a and R3b), which were homogeneous static isotropic scattering media²⁴. We experimentally measured the anisotropic factor $g \approx 0.02$, and the scattering coefficient $\mu_s \approx 0.34 \text{ mm}^{-1}$. An object (the same as Fig. R2a) was sandwiched in the polyethylene foams. The incoherent light transmitted through a polyethylene foam of 5 mm thickness, the object and a polyethylene foam of 10 mm thickness in turn. Then, speckle images formed on the back surface of the 10 mm thick polyethylene foam were recorded during the motion of the object (Fig. R3c). Based on these speckle images, the object's autocorrelation was constructed (Fig. R3d). Then, the object image was reconstructed (Fig. R3e). However, when we replaced the 5 mm thick polyethylene foam with a 10

mm thick polyethylene foam and kept the other conditions unchanged, the speckle contrast became too low to image although the entire envelope was measured.

According to the experiment results, the maximum imaging depth here is between 15 mm to 20 mm thicknesses of polyethylene foams, corresponding to $5.1\sim 6.8 l_s$ and $5.0\sim 6.7 l_t$. Notably, the results of maximum imaging depth in polyethylene foam ($5.0\sim 6.7 l_t$) and parafilm ($6.2 l_t$) are consistent after eliminating the anisotropic effects of the samples. Therefore, we conclude that the imaging depth limit of speckle kinetography is about $6 l_t$.

Action taken

In the revised manuscript, all the descriptions of “imaging in scattering media of arbitrary thickness” have been revised as “imaging in scattering media with thicknesses up to about 6 transport mean free paths”.

In order to discuss the imaging depth limit, the first paragraph of the Discussion in the manuscript has been revised as follows:

“To explore the imaging depth limit of speckle kinetography, imaging of an object consisting of three $25\ \mu\text{m}$ width lines moving inside parafilm of 2 to 30 layers is experimentally implemented. The structural similarity index measurement (SSIM) of the constructed autocorrelation decreases as the sample thickness L increases (Supplementary Fig. 6a). When the sample thickness reaches 28 layers, the object’s autocorrelation becomes too blurry to image. Therefore, the maximum imaging depth here is about 26 layers of parafilm, corresponding to $19.9 l_s$ and $6.2 l_t$. To determine the main limiting factors, the envelope expansion, the relative displacement error and the speckle contrast versus the sample thickness are analyzed. The full width at half-maximum W of the envelope broadens as the sample thickness L increases (Supplementary Fig. 6b). But the entire envelope can still be measured when the parafilm is 28 layers, so it does not have obvious impact on imaging. The relative displacement error E fluctuates slightly as L increases (Supplementary Fig. 6c). But the fluctuation keeps within ± 4 camera pixels, which is smaller than the pixel size of the object’s autocorrelation, corresponding to 16 camera pixels in these experiments. Thus, it does not cause misplacement to autocorrelation pixels. The speckle contrast decreases as L increases and it quickly drops below 0.1 from 26 to 30 layers (Supplementary Fig. 6d). The low contrast causes the speckles to overlap with each other even when the corresponding objects do not overlap. In this case, the small values of the object’s autocorrelation become large, which is consistent with the blurred autocorrelation constructed at 28 layers (Supplementary Fig. 6a). It causes the imaging to fail. In conclusion, although the above three factors all limit the imaging depth, the speckle

contrast dominates.

Supplementary Fig. 6 The influence of medium thickness on four factors. Imaging of an object consisting of three 25 μm width lines moved in parafilm of 2 to 30 layers is experimentally performed. **a**, The SSIM of the constructed autocorrelation versus the sample thickness L . **b**, The FWHM W of the envelopes versus L . **c**, The relative displacement error E versus L . **d**, The speckle contrast versus L .

To eliminate the anisotropic effects of the scattering sample, imaging of the above-mentioned 25 μm widths object embedded in homogeneous static isotropic polyethylene foams²⁴ (Supplementary Figs. 7a and 7b) is experimentally performed. When the polyethylene foam is 15 mm thick, the speckle images with entire envelope and sufficient speckle contrast are recorded under narrowband and broadband illumination (Supplementary Figs. 7c and 7d). The constructed autocorrelations and recovered object images are shown in Supplementary Figs. 7e-7h. However, when the polyethylene foam is 20 mm thick, the contrast of the speckle images is too low to image. Therefore, the maximum imaging depth here is between 15 mm to 20 mm thicknesses of polyethylene foams, corresponding to 5.1~6.8 l_s and 5.0~6.7 l_t . Notably, the maximum imaging depths in polyethylene foam (5.0~6.7 l_t) and parafilm (6.2 l_t) are consistent after eliminating the anisotropic effects. Therefore, we conclude that the imaging depth limit of speckle kinetography is about 6 l_t .

Supplementary Fig. 7 Imaging of an object embedded in 15 mm thick polyethylene foams. **a** and **b**, Front (**a**) and side (**b**) photographs of the polyethylene foams. **c**, One of the recorded speckle images under narrowband illumination. **d**, One of the recorded speckle images under broadband illumination. **e**, The object’s autocorrelation constructed from the recorded speckle images in (**c**). **f**, The object’s autocorrelation constructed from the recorded speckle images in (**d**). **g**, The object image recovered from (**e**). **h**, The object image recovered from (**f**). Scale bars: 50 μm .”

The methods and experiments for measuring the anisotropy coefficient g of the polyethylene foams and parafilm have been added in Supplementary Note 10.

3. Reviewer comment

In Eq. (3), the authors assumed that “the impulse responses do not overlap with each other”. However, because $s(x, y)$ is low-contrast speckles with a positive mean value, the orthogonality between $s(x', y')$ and $s(x, y)$ in Eq. (8) does not hold. This could make the autocorrelation inaccurate, making phase retrieval unstable. Considering that phase retrieval is highly sensitive to the auto-correlation, it is unclear how this noise term can be easily neglected without additional processing.

Response

We thank the reviewer for pointing out this concern. This assumption is made to get a strict derivation of Eq. (3). In practice, the Hadamard product of $s(\xi, \eta; x', y')$ and $s(\xi, \eta; x, y)$ is greater than zero when the speckle contrast is low, which usually occurs in the conditions of thick media or broadband illumination. In this case, these positive values contribute to the background term $B_{ij}(\Delta x_{ij}, \Delta y_{ij})$ in Eq. (8) (it has been changed as Eq. (9) in the revised manuscript), which causes the intensity fluctuation of the object’s autocorrelation. The term $B_{ij}(\Delta x_{ij}, \Delta y_{ij})$ is usually in a bell shape, like the autocorrelation of a halo. Therefore, the autocorrelation constructed from low-contrast speckles often likes an object’s autocorrelation fluctuating on the surface of a bell-shaped distribution. Since the background term $B_{ij}(\Delta x_{ij}, \Delta y_{ij})$ only slightly reduces the contrast rather than changes the structure of the object’s autocorrelation, it has limited impact on the phase retrieval algorithm with constraints of realness and non-negativity.

We would like to explain the influence of speckle contrast on imaging through a group of comparative experiments. As mentioned in the above response, the speckle contrast decreases as the sample thickness increases. Based on this, experiments for imaging of a 25 μm width object moving inside parafilm with thicknesses of 2, 10 and 20 layers are implemented. The corresponding object's autocorrelations are constructed and shown in Figs. R4a-R4c. As the speckle contrast decreases, a bell-shaped background term occurs (Fig. R4d and the inset). The slope of the bell-shaped distribution increases with the decrease of speckle contrast, refer to the curves B_1 and B_2 in Fig. R4d, which are inferred from the intensity distributions along the x direction of the autocorrelations (dashed lines in Figs. R4a-R4c).

Nevertheless, the object images recovered from these autocorrelations are not greatly affected, as shown in Figs. R4e-R4g. The SSIMs of these object images are respectively 0.81, 0.83 and 0.82. In all experiments, the phase retrieval algorithm follows Katz et al.^{13,28} (also described below in detail) and there is no additional processing. By the way, additional processing to remove a bell-shaped distribution from the constructed autocorrelation may be a possible way to further improve the imaging quality.

Fig. R4 Imaging within parafilm of 2, 10 and 20 layers. **a-c,** The object's autocorrelations constructed from corresponding sample thicknesses. **d,** Intensity distributions along the x direction of the autocorrelations that are marked with dashed lines in (a-c). The curves B_1 and B_2 are inferred from the corresponding intensity distributions that affected by the bell-shaped background term. Its two-dimensional shape can refer to the inset. **e-g,** The object images directly recovered from the corresponding autocorrelations through the phase retrieval algorithm^{13,28} without additional processing. Scale bars: 50 μm .

Action taken

In the Methods of the revised manuscript, we have added a paragraph of the related explanation below Eq. (9) as follows:

“Indeed, $B_{ij}(\Delta x_{ij}, \Delta y_{ij})$ occurs not only when the illumination is broadband, but also when the scattering medium is optically thick. The root cause is the decrease of speckle contrast in these conditions. $B_{ij}(\Delta x_{ij}, \Delta y_{ij})$ is usually a bell shape, like the autocorrelation

of a halo. Therefore, the autocorrelation constructed from low-contrast speckles often looks like an object's autocorrelation fluctuating on the surface of a bell-shaped distribution. And the slope of the bell-shaped distribution increases with the decrease of speckle contrast. Thus, the contrast of the constructed autocorrelation decreases with the speckle contrast. Nevertheless, the object images recovered through the phase-retrieval algorithm^{13,28} are little affected, because the low-contrast speckle only slightly affects the contrast rather than the structure of the object's autocorrelation, which has limited impact on the phase retrieval algorithm with constraints of realness and non-negativity (Supplementary Note 13).”

4. Reviewer comment

Related to my previous comment, the normalized speckle S is obtained by dividing the speckle image with its envelope. The odd thing is that the values outside the envelope have near 0 values. This will make S have noisy values outside the envelope. However, in Extended Fig. 2, S also shows near 0 values outside the envelope. This part needs more explanation.

Response

We thank the reviewer for pointing out this concern. As shown in Fig. R5a, the values of a speckle image are in the range of 0 to 153. The values along the middle row (solid blue line in Fig. R5a) are also shown as the solid blue curve in Fig. R5e. The low-pass filtered envelope is shown in Fig. R5b. The values along the middle row (orange dashed line in Fig. R5b) are also shown as the orange dashed curve in Fig. R5e. It can be seen that the distribution of the envelope follows the general distribution of the speckle image, but the high-frequency details have been filtered. The values at the edges of the speckle image and the envelope are approximately equal and positive. Therefore, by dividing the speckle image with its envelope, the spatially normalized speckle S (Fig. R5c) fluctuates around 1, which can be more directly observed in Fig. R5e (red dotted curves in Fig. R5e and the inset). In order to show the normalized speckle S in Fig. R5c more clearly, the colour bar range is changed from 0~19.8 to 0.95~2.77 (Fig. R5d). Owing to the difference of colour bar ranges, the positive values at the edges of the normalized speckle S and the envelope both look like near 0.

Fig. R5 Generation of envelope and speckle from a speckle image. **a**, A speckle image. **b**, The low-pass filtered envelope. **c**, The spatially normalized speckle S . **d**, The spatially normalized speckle S shown under a colour bar of a small range. **e**, The values along the middle row of images (**a-d**). The solid blue curve represents the distribution of the speckle image. The orange dashed curve represents the distribution of the envelope. The red dotted curve represents the distribution of the spatially normalized speckle S , which is also shown in the inset.

Action taken

To give the readers a clear understanding about the generation of envelope and speckle from a speckle image, the above-mentioned explanations have been added in the Supplementary Note 11 of the manuscript.

5. Reviewer comment

There are other techniques to image/track hidden objects such as [Applied optics, 57(4), 905-913., IEEE Photonics Journal, 11(6), 1-14., Nature communications, 13(1), 5779.]. The authors should discuss the difference between their work and the existing studies.

Response

We thank the reviewer for this helpful suggestion. In fact, the tracking accuracy of these works is better than our proposed method. However, a prerequisite of these works is that the cross-correlation of objects before and after the movements must be obtained from the correlation of their corresponding speckles. Thus, the memory effect is an essential condition of these works for imaging and tracking of the hidden objects. As a result, the application of these methods is limited in some scenarios that the angular memory effect range is extremely small, such as the scattering medium is optically thick or the object is very close to the medium. In these cases, speckle kinetography can cope

with these scenarios as an alternative.

For example, a moving object consisting of three 25 μm width lines is embedded in parafilm of 6 layers. One of the recorded speckle images is shown in Fig. R6a. A part of the trajectory and the object image recovered through speckle kinetography are shown in Fig. R6b. Compared with the real trajectory (solid red line in Fig. R6b), the recovered trajectory (yellow points in Fig. R6b) have errors within 3 camera pixels. But the errors do not cause deformation of the autocorrelation since a pixel of the constructed autocorrelation corresponds to 16 camera pixels in this experiment. Nevertheless, the autocorrelation of the speckle (Fig. R6c) does not contain the object's autocorrelation. That is, the memory effect range in this object plane is smaller than the object size. Moreover, the cross-correlations of the speckles from objects with various displacements do not contain the cross-correlation of objects before and after the movements, as shown in Figs. R6d-R6i. In this case, imaging or tracking of objects cannot be achieved by these existing methods.

Fig. R6 Imaging of a moving object embedded in parafilm of 6 layers. **a**, One of the recorded speckle images. **b**, A part of the trajectory and the object image recovered through speckle kinetography. The object consists of three 25 μm width lines. The yellow points are the recovered trajectory. The solid red line represents the real trajectory. **c**, The autocorrelation of the speckle. **d-i**, The cross-correlations of the speckles from objects before and after moving with distances of 20 μm , 40 μm , 60 μm , 80 μm , 100 μm and 125 μm . The speckles' correlations do not contain the object's autocorrelation.

Action taken

In the revised manuscript, discussion of the difference between the existing studies and our work have been added in the fifth paragraph of the Discussion as follows:

“The relative displacements recovered from the envelopes are not so accurate as the existing methods³⁰⁻³². However, a prerequisite of these methods is that the cross-correlation of objects before and after the movements must be obtained from the correlation of speckles. Therefore, the memory effect is an essential condition of these

methods. As a result, the application of these methods is limited in some scenarios that the angular memory effect range is very small, such as the scattering medium is optically thick or the object is very close to the medium. In these cases, speckle kinetography can cope with these scenarios as an alternative.”

6. Reviewer comment

The term “speckle masking” had already been used for the bispectrum analysis of speckles [Applied Optics, 22(24), 4028-4037.]. Using the term for this work can be confusing since there is non-invasive imaging of moving objects based on the bispectrum analysis [IEEE Photonics Journal, 11(6), 1-14.].

Response

We thank the reviewer for this valuable suggestion. We would like to use “overlapping speckle” instead of “speckle masking” after careful thought. Because the result of the Hadamard product between speckles only retains the overlapping speckle between them, “overlapping speckle” is more perspicuous than “speckle masking”.

Action taken

In the revised manuscript, the sentences around Eq. (3) have been revised as follows: “Under this assumption, the overlapping speckle between these two speckles is extracted through a Hadamard product as follows:

$$\begin{aligned} M_{ij}(\xi, \eta; \Delta x_{ij}, \Delta y_{ij}) &= S_i(\xi, \eta) S_j(\xi, \eta; \Delta x_{ij}, \Delta y_{ij}) \\ &= \iint O(x, y) O(x - \Delta x_{ij}, y - \Delta y_{ij}) I_s^2(x, y) s^2(\xi, \eta; x, y) dx dy. \end{aligned} \quad (3)$$

The result only retains impulse responses from the point sources within the overlapping region of the object before and after the motion displacement $(\Delta x_{ij}, \Delta y_{ij})$.”

In addition, all the terms “speckle masking” have been changed with “overlapping speckle” throughout the manuscript.

7. Reviewer comment

For the broadband experiment, the authors claimed that the proposed method can avoid “image broadening problem”. But it is unclear why an object imaged without a scattering medium is blurry, as in Fig. 3(d). The broadening appears to be due to the chromatic aberration of optics, which can be easily avoided with proper lenses. Furthermore, in lines 253-255, the authors described that the proposed method works without a medium. However, in such a case, the direct image of an object can be easily obtained. I am not sure if these are important points to discuss.

Response

We thank the reviewer for pointing out this concern. In fact, the lens we used in all experiments is an achromatic lens with anti-reflective coating at 400-750 nm. Its focal length specification wavelengths are 486.1 nm, 587.6 nm and 656.3 nm. However, the spectral range of the white light source is about 260-2500 nm (dotted curve marked as ‘6255’ in Fig. R7a), and the response range of the camera is about 350-1000 nm (Fig. R7b). These wider wavelength ranges cause the direct imaging results to suffer from broadening problem. Nevertheless, although using the same lens, light source and camera as direct imaging, the scattering imaging results of speckle kinetography are not broadened by dispersion. These experimental results of imaging with and without scattering media can be used as a strong proof that the speckle kinetography can avoid the imaging broadening problem.

Fig. R7 The wavelength ranges of the white light source and the camera used in the experiments. a, The spectral range of the research arc lamp source. **b,** The response range of the camera.

Action taken

As the previous description about the dispersion of direct imaging was not clear enough for the reader, we have revised the last sentence of the first paragraph in “Imaging under broadband illumination” section of the Results as follows:

“In contrast, although the same achromatic lens as the scattering imaging was used, the direct imaging results in Fig. 3 were strongly broadened due to dispersion of the white light.”

In addition, the dispersion-related parameters have been added in the “Experimental setup” section in the Methods of the revised manuscript as “an achromatic lens (75 mm focal length, 50.8 mm diameter, anti-reflective coating at 400-750 nm, focal length specification wavelengths at 486.1 nm, 587.6 nm and 656.3 nm) and a monochrome camera (5488×3672 pixels, 2.4 μm pixel size, 350 to 1000 nm response range, dynamic range of 8 bits)”.

Moreover, the above-mentioned descriptions about the dispersion of direct imaging as well as the figure of wavelength ranges have been added in the Supplementary Note 12 and Supplementary Fig. 13 of the revised manuscript.

Besides, in the revised manuscript, all the descriptions about the availability of imaging without scattering medium, including the descriptions in lines 253-255, have been removed since it is apparent and not important.

8. Reviewer comment

It is not clear which phase retrieval algorithm was used. Fienup-type phase retrieval can be any algorithm that employs an object plane constraint. The authors need to explain this in more detail.

Response

We thank the reviewer for pointing out this ambiguity. The iterative Fienup-type phase-retrieval algorithm we used follows Katz et al.¹³ [Nature Photonics, 8, 784-790 (2014)]. The algorithm flow chart is given in Fig. R8. This modified Gerchberg-Saxton algorithm starts with an initial guess for the object pattern $o_1(x, y)$. This initial guess, chosen as a random pattern, is entered to the algorithm that performs the following 4 steps at its i^{th} iteration:

$$\begin{aligned} O_i(k_x, k_y) &= FT\{o_i(x, y)\}, \\ \theta_i(k_x, k_y) &= \arg\{O_i(k_x, k_y)\}, \\ O'_i(k_x, k_y) &= \sqrt{|FT\{C(x, y)\}|} e^{j\theta_i(k_x, k_y)}, \\ o'_i(x, y) &= FT^{-1}\{O'_i(k_x, k_y)\}, \end{aligned}$$

where the $C(x, y)$ in the 3rd step is the measured object's autocorrelation.

The input for the next ($i+1$) iteration, $o_{i+1}(x, y)$, is obtained from the output of the i^{th} iteration, $o'_i(x, y)$, by imposing physical constraints on the object image, which is real and non-negative in our implementation. Two types of implementations of these constraints are used in the algorithm^{13,14,28}, termed the ‘Hybrid Input-Output (HIO)’ algorithm:

$$o_{i+1}(x, y) = \begin{cases} o'_i(x, y) & \text{for } (x, y) \notin \Gamma \\ o_i(x, y) - \beta o'_i(x, y) & \text{for } (x, y) \in \Gamma \end{cases},$$

and the ‘Error reduction’ algorithm:

$$o_{i+1}(x, y) = \begin{cases} o'_i(x, y) & \text{for } (x, y) \notin \Gamma \\ 0 & \text{for } (x, y) \in \Gamma \end{cases},$$

where Γ is the set of all points (x, y) on $o'_i(x, y)$ that violate the physical constraints, and

β is a feedback parameter that control the convergence properties of the algorithm.

Fig. R8 Flow chart of the iterative phase-retrieval algorithm used. The Fienup’s HIO phase-retrieval algorithm and the Fienup’s error-reduction algorithm are used^{13,14,28}. Both algorithms are based on an iterative modified Gerchberg-Saxton algorithm. The $C(x, y)$ is the experimentally measured object’s autocorrelation.

Following Katz et al.¹³, a few thousand iterations of the HIO algorithm were ran with a decreasing beta factor from $\beta=2$ to $\beta=0$, in steps of 0.04. For each β value, 40 iterations of the algorithm were performed. The result of the HIO algorithm was fed as an input to additional 40 iterations of the ‘Error reduction’ algorithm to obtain the final result.

To assure faithful reconstruction of each image, several different runs of the algorithm (from 10 up to 60, typically 20) were performed with different random initial conditions, and the reconstruction having the closest Fourier spectrum to the measured autocorrelation Fourier transform (lowest mean-square-error) was chosen as the final reconstructed result.

Action taken

In the revised manuscript, the description about the phase retrieval algorithm used has been added in the Supplementary Note 13.

9. Reviewer comment

In line 216, it is unclear what “edge value” and “the excess edge can be disregarded” means.

Response

We thank the reviewer for pointing out this ambiguity. As mentioned in the above responses, the distribution of $B_{ij}(\Delta x_{ij}, \Delta y_{ij})$ is usually a bell shape. Therefore, the values at the edge of the bell-shaped $B_{ij}(\Delta x_{ij}, \Delta y_{ij})$ are small. Compared with the object’s autocorrelation, the overall size of $B_{ij}(\Delta x_{ij}, \Delta y_{ij})$ is slightly larger. Owing to the small values distributed at the edge of $B_{ij}(\Delta x_{ij}, \Delta y_{ij})$, the values at the edges beyond the autocorrelation range have limited effect on the overall size of the object’s autocorrelation.

Action taken

In the revised manuscript, this sentence has been rewritten as follows:

“Compared with the object’s autocorrelation, the pixel size of $\Sigma B_{ij}(\Delta x_{ij}, \Delta y_{ij})$ is the same, but the overall size is slightly larger. Since the values at the edge of $\Sigma B_{ij}(\Delta x_{ij}, \Delta y_{ij})$ are usually small (see Methods), those values beyond the autocorrelation range have limited effect on the overall size of the object’s autocorrelation.”

10. Reviewer comment

The authors should include the NAs of objective lenses used.

Response and action taken

We thank the reviewer for reminding us of the parameter information missing. In the revised manuscript, the NAs of objective lenses have been added in the “Experimental setup” section of the Methods as: “an objective (40×, NA 0.65 for thin media or 63×, NA 0.85 for thick media)”.

11. Reviewer comment

Regarding Eq. (5), I suggest the authors to rewrite the autocorrelation to $O^*O(\Delta x)$ instead of $O(\Delta x)^*O(\Delta x)$.

Response and action taken

We thank the reviewer for this professional suggestion. In the revised manuscript, we have rewritten the autocorrelation to $O^*O(\Delta x)$ in Eqs. (5), (6) and (9).

Reviewer 2:

1. Reviewer comment

In this manuscript the authors propose a new approach in the family of speckle autocorrelation imaging techniques to visualize objects that are hidden within a scattering medium. Originally proposed by Katz et al. in Nature Photonics (2014) and Bertolotti et al. in Nature (2012), this category of techniques utilizes speckle patterns to reconstruct an object’s 2D autocorrelation. Images are then recovered using a phase-retrieval algorithm.

In the present work, authors demonstrate the utility of this approach for objects that are moving inside the scattering medium and that surprisingly few recordings are needed to sample the objects’ autocorrelation. To acquire the raw data, they move the sample along at least two axes (e.g. a T-shaped or U-shaped pattern) and record speckle pattern for each location. To recover the object’s 2D autocorrelation, they use low-pass filtering to estimate the relative shift between two recordings and they calculate the dot product between the high-pass filtered speckle patterns to get the value of the

autocorrelation at this shift. Once an object's 2D autocorrelation is obtained, they use phase-retrieval to recover the image.

I find this work to be conceptually interesting and relevant to the field of imaging in scattering media. The description of the experiments and data analysis is clear and should make the paper easy to follow and reproduce. However, the authors should address the following issues before I can support publication:

Major comments:

Starting with the abstract and in multiple places in the manuscript, the authors claim that their technique allows imaging inside scattering media of arbitrary (unlimited) thickness with diffraction-limited resolution. However, this method critically depends on two factors that degrade with medium thickness: first, an accurate estimation of the relative shift of the envelopes and second, speckle contrast. These limitations should be discussed in detail and the manuscript's claims should be adjusted accordingly.

Response

We thank the reviewer for the professional review and approval of our work. In order to explore the imaging depth limit of speckle kinetography, a group of experiments was implemented.

Fig. R9 The influence of medium thickness on three factors. **a**, A hidden object consisting of three 25 μm width lines. **b**, The SSIM of the constructed autocorrelation versus the sample thickness L . **c**, The relative displacement error E versus L . **d**, The speckle contrast versus L . Scale bar: 50 μm .

In the experiments, the scattering samples were parafilm of 2 to 30 layers, in steps of 2 layers. As shown in Fig. R9a, a transmissive object consisting of three lines with 25 μm widths and 125 μm height was embedded in the middle of the parafilm. This object was larger than the objects described in the manuscript so that sufficient light intensity could be detected even when the parafilm was thicker than 22 layers. In each sample thickness, 52 speckle images from this axisymmetric object moved in an L shape were recorded, in steps of 5 μm . According to the recorded 780 speckle images, three curves were calculated (Figs. R9b-R9d).

The first curve (Fig. R9b) shows that the structural similarity index measurement

(SSIM) of the constructed autocorrelation decreases as the sample thickness L increases. The downward trend of SSIM is relatively slow from 2 to 26 layers, and turns steep from 26 to 30 layers. When the sample thickness reaches 28 layers, the object's autocorrelation is too blurry to recover the correct object image. Therefore, the maximum imaging depth here is about 26 layers of parafilm, corresponding to 19.9 scattering mean free path l_s and 6.2 transport mean free path l_t , which is calculated from the experimentally measured anisotropy coefficient g of 0.69.

Next, we would like to determine the main limiting factors between the relative displacement error and the speckle contrast. The curve in Fig. R9c shows that the relative displacement error E fluctuates slightly as L increases. And the fluctuation keeps within ± 4 camera pixels. Since the pixel size of the constructed object's autocorrelation corresponds to 16 camera pixels in these experiments, fluctuations within ± 8 camera pixels do not cause misplacement to autocorrelation pixels. If the fluctuation exceeds half autocorrelation pixel, it will cause deformation of the object's autocorrelation. Owing to the limited fluctuation of E , the relative displacement error is not the main limitation here. The curve in Fig. R9d shows that the speckle contrast decreases as the sample thickness L increases. Moreover, the speckle contrast quickly drops below 0.1 from 26 to 30 layers. Owing to the extremely low contrast, the speckles overlap with each other even when the corresponding objects do not overlap. It will cause the small values in the object's autocorrelation to become large, which is consistent with the blurred autocorrelation constructed at 28 layers (Fig. R9b). In conclusion, both the relative displacement error and the speckle contrast decrease limit the imaging depth, and the speckle contrast decrease dominates.

Action taken

In order to discuss the imaging depth limit, the first paragraph of the Discussion in the manuscript has been revised as follows:

“To explore the imaging depth limit of speckle kinetography, imaging of an object consisting of three 25 μm width lines moving inside parafilm of 2 to 30 layers is experimentally implemented. The structural similarity index measurement (SSIM) of the constructed autocorrelation decreases as the sample thickness L increases (Supplementary Fig. 6a). When the sample thickness reaches 28 layers, the object's autocorrelation becomes too blurry to image. Therefore, the maximum imaging depth here is about 26 layers of parafilm, corresponding to 19.9 l_s and 6.2 l_t . To determine the main limiting factors, the envelope expansion, the relative displacement error and the speckle contrast versus the sample thickness are analyzed. The full width at half-maximum W of the envelope broadens as the sample thickness L increases

(Supplementary Fig. 6b). But the entire envelope can still be measured when the parafilm is 28 layers, so it does not have obvious impact on imaging. The relative displacement error E fluctuates slightly as L increases (Supplementary Fig. 6c). But the fluctuation keeps within ± 4 camera pixels, which is smaller than the pixel size of the object's autocorrelation, corresponding to 16 camera pixels in these experiments. Thus, it does not cause misplacement to autocorrelation pixels. The speckle contrast decreases as L increases and it quickly drops below 0.1 from 26 to 30 layers (Supplementary Fig. 6d). The low contrast causes the speckles to overlap with each other even when the corresponding objects do not overlap. In this case, the small values of the object's autocorrelation become large, which is consistent with the blurred autocorrelation constructed at 28 layers (Supplementary Fig. 6a). It causes the imaging to fail. In conclusion, although the above three factors all limit the imaging depth, the speckle contrast dominates.

Supplementary Fig. 6 The influence of medium thickness on four factors. Imaging of an object consisting of three $25 \mu\text{m}$ width lines moved in parafilm of 2 to 30 layers is experimentally performed. **a**, The SSIM of the constructed autocorrelation versus the sample thickness L . **b**, The FWHM W of the envelopes versus L . **c**, The relative displacement error E versus L . **d**, The speckle contrast versus L .”

2. Reviewer comment

Related: How does sample thickness affect the precision with which the shifts of the object autocorrelation function are reconstructed? Given that the authors claim that the

tissue thickness did not visibly affect the object reconstruct, could it be attributed to the scattering anisotropy of the samples (parafilm)? In this case, the transport mean free path of the sample would be much greater than the scattering mean free path, and thus the memory effect will be present at much greater depths. Would this method still work when using a strong diffuser? In a similar fashion, strongly diffusive samples would exhibit a much stronger spectral broadening, blurring the resulting speckle and making the reconstruction for broadband illumination impossible.

Response

We thank the reviewer for the valuable suggestions. As mentioned above, the imaging depth limit is about $6.2 l_s$. On this basis, two further studies, including ‘the memory effect range’ and ‘imaging in isotropic scattering media’, are implemented.

Fig. R10 The influence of memory effect range on the autocorrelation of speckles. An object “5” that stands tens of centimeters away from the scattering medium is illuminated by a pseudo-thermal spatially-incoherent source. The object light is scattered and recorded by a camera. **a-c**, The recorded speckles that are scattered by the scattering medium with optical thickness of $2.1 l_s$ (**a**), $4.9 l_s$ (**b**) and $9.1 l_s$ (**c**). **d-f**, The range of object’s autocorrelation in the autocorrelation of the corresponding speckle (yellow dotted circle). It becomes smaller since the memory effect range decreases with the increase of medium thickness.

As known, the angular memory effect range decreases with the increase of optical thickness of scattering media (Figs. R10a-R10c). Accordingly, the range of the object’s autocorrelation acquired from the autocorrelation of the speckle will become smaller (Figs. R10d-R10f). Therefore, a direct way to determine the memory effect range is observing the range of the object’s autocorrelation from the autocorrelation of the speckle. Figure R11 shows the autocorrelation of the speckles recorded in our experiments. As the objects are embedded in rather than tens of centimeters away from the scattering media, the memory effect range is very small at this object plane. Only the autocorrelation of the speckle generated from the $1 \mu\text{m}$ width object hidden in 6 layers of parafilm (Fig. R11b) shows obvious object’s autocorrelation. The autocorrelations of the speckles from the $1 \mu\text{m}$ width object in 14 layers of parafilm

(Fig. R11c) and the 10 μm width object in 6 and 14 layers of parafilm (Figs. R11f and R11g) show a very small part of object's autocorrelation. The object's autocorrelation is hard to observe from the autocorrelations of the rest speckles in Fig. R11. In contrast, all the objects correspond to the speckles in Fig. R11 can be imaged via speckle kinetography. Therefore, the memory effect makes little difference to speckle kinetography.

Fig. R11 Autocorrelation of speckles. **a**, An object consists of six 1 μm widths lines. **b-d**, Autocorrelations of speckles generated from the object in **(a)** embedded in parafilm of 6 **(b)**, 14 **(c)** and 22 **(d)** layers. **e**, An object consists of two 10 μm widths lines. **f-h**, Autocorrelations of speckles generated from the object in **(e)** embedded in parafilm of 6 **(f)**, 14 **(g)** and 22 **(h)** layers. **i**, An object consists of three 25 μm widths lines. **j-l**, Autocorrelations of speckles generated from the object in **(i)** embedded in parafilm of 6 **(j)**, 14 **(k)** and 22 **(l)** layers. Scale bars: 10 μm .

In order to eliminate the influence of the anisotropic effect of the samples, experiments for imaging of the above-mentioned 25 μm width object moving inside isotropic samples under narrowband and broadband illumination are implemented. The scattering samples were polyethylene foams (Figs. R12a and R12b), which were homogeneous static isotropic scattering media²⁴. We experimentally measured the anisotropic factor $g \approx 0.02$ and the scattering coefficient $\mu_s \approx 0.34 \text{ mm}^{-1}$. The object was embedded in the polyethylene foams. The incoherent light transmits through a 5 mm thick polyethylene foam, the object and a 10 mm thick polyethylene foam in turn. Then, speckle images formed on the back surface of the 10 mm thick polyethylene foam were recorded during the motion of the object (Fig. R12c for narrowband illumination and Fig. 12f for broadband illumination). Based on these speckle images, the object's autocorrelations were constructed (Figs. R12d and R12g). Then, the object images were reconstructed (Figs. R12e and R12h). However, when we replaced the 5 mm thick polyethylene foam with a 10 mm thick polyethylene foam and kept the other conditions unchanged, the speckle contrast became too low to image.

Fig. R12 Imaging of an object embedded in 15 mm thick polyethylene foams. **a** and **b**, Front (**a**) and side (**b**) photographs of the polyethylene foams. **c**, One of the recorded speckle images under narrowband illumination. **d**, The object's autocorrelation constructed from the recorded speckle images in (**c**). **e**, The object image recovered from (**d**). **f**, One of the recorded speckle images under broadband illumination. **g**, The object's autocorrelation constructed from the recorded speckle images in (**f**). **h**, The object image recovered from (**g**). Scale bars: 50 μm .

According to the experiment results, the maximum imaging depth here is between 15 mm to 20 mm thicknesses of polyethylene foams, corresponding to $5.1\sim 6.8 l_s$ and $5.0\sim 6.7 l_t$. Notably, the results of maximum imaging depth in polyethylene foam ($5.0\sim 6.7 l_t$) and parafilm ($6.2 l_t$) are consistent after eliminating the anisotropic effects of the samples. Therefore, we conclude that the imaging depth limit of speckle kinetography is about $6 l_t$.

Action taken

We have added the results of imaging in isotropic scattering media in the second paragraph of the Discussion as follows:

“To eliminate the anisotropic effects of the scattering sample, imaging of the above-mentioned 25 μm widths object embedded in homogeneous static isotropic polyethylene foams²⁴ (Supplementary Figs. 7a and 7b) is experimentally performed. When the polyethylene foam is 15 mm thick, the speckle images with entire envelope and sufficient speckle contrast are recorded under narrowband and broadband illumination (Supplementary Figs. 7c and 7d). The constructed autocorrelations and recovered object images are shown in Supplementary Figs. 7e-7h. However, when the polyethylene foam is 20 mm thick, the contrast of the speckle images is too low to image. Therefore, the maximum imaging depth here is between 15 mm to 20 mm thicknesses of polyethylene foams, corresponding to $5.1\sim 6.8 l_s$ and $5.0\sim 6.7 l_t$. Notably, the maximum imaging depths in polyethylene foam ($5.0\sim 6.7 l_t$) and parafilm ($6.2 l_t$) are consistent after eliminating the anisotropic effects. Therefore, we conclude that the imaging depth limit of speckle kinetography is about $6 l_t$.” The Supplementary Fig. 7 mentioned in this paragraph is the same as Fig. R12.

Besides, the methods and experiments for measuring the anisotropy coefficient g of

the polyethylene foams and parafilm have been added in the Supplementary Note 10.

3. Reviewer comment

Minor comments:

Please include the example recorded intensity images in Figs. 1 and 2 next to the reconstructed autocorrelations.

Response and action taken

We thank the reviewer for this helpful suggestion. In the revised manuscript, the example recorded speckle images have been added in Figs. 1-3 as follows:

Fig. 1 Schematic of the principle, apparatus and computational model for speckle kinetography. **a**, Principle. Under incoherent illumination, the intensity distribution on object plane is $I_S(x, y)$. Every point source forms an impulse response $s(\xi, \eta; x, y)$ on the back surface of scattering medium. The recorded speckle image I is the superposition of the impulse responses generated within the object. The information of overlap C and relative position $(\Delta x, \Delta y)$ between any two motion states of an object is retained in the high-pass filtered speckles S and low-pass filtered envelopes E . The information is fully utilized to construct the object's autocorrelation $C(\Delta x, \Delta y)$ for imaging. **b**, Experimental setup. A movable object is embedded in an unknown scattering medium, which can be optically thick scattering media or thin scattering layers. A simple and speckle pattern-sized trajectory is sufficient for imaging. The front surface of the medium is illuminated via an incoherent light source, which can either be narrowband or broadband, and speckles are formed at the back surface of the medium. A series of magnified speckle images I are noninvasively detected via a simple imaging system consisting of a lens and a monochrome camera. **c**, Computational model. The speckle images are numerically separated into spatially normalized speckles S and slowly varying envelopes E . The overlap C is calculated from the sum of the overlapping speckle M , which is sifted from any two of the speckles by Hadamard product. The relative position $(\Delta x, \Delta y)$ is calculated from the corresponding two envelopes via cross-correlation. Combining all the extracted information, the object's autocorrelation is constructed and then utilized for object image reconstruction.

Fig. 2 Experimental imaging within thin to thick parafilm under single-colour LED illumination. **a-d**, Movable objects consisting of two 10 μm width lines (**a**), three 5 μm width lines (**b**), six 1 μm width lines (**c**) and a number-shaped object with a height of 100 μm (**d**) are embedded in parafilm of 6, 14, 22 and 0 layers. One of the recorded speckle images, the constructed autocorrelation and the reconstructed image are successively shown in each inset. Scale bars: 10 μm .

Fig. 3 Experimental imaging within thin to thick parafilm under white light illumination. **a-d**, Movable objects consisting of two 10 μm width lines (**a**), three 5 μm width lines (**b**), six 1 μm width lines (**c**) and a number-shaped object with a height of 100 μm (**d**) are embedded in parafilm of 6, 14, 22 and 0 layers. One of the recorded speckle images, the constructed autocorrelation and the reconstructed image are successively shown in each inset. The imaging broadening problem is avoided because the space invariance is released in speckle kinetography. The imaging results without scattering medium in the path suffer from a broadening problem due to dispersion of white light. Scale bars: 10 μm .

4. Reviewer comment

Please remove the sentence “Interestingly, speckle kinetography works even when there is no medium in the optical path because the impulse response degenerates to a point. In a sense, the effect of the scattering medium on imaging is eliminated.” from the Discussion as this is self-evident.

Response and action taken

We thank the reviewer for this valuable suggestion. In the revised manuscript, the sentence about the availability of imaging without scattering medium has been removed. In addition, the related discussions in the manuscript including that in Fig. 1 have been deleted.

5. Reviewer comment

Reconstructed trajectories in Extended Fig. 5 are hardly visible. Could the authors plot them more visibly and add the real trajectories for comparison?

Response

We thank the reviewer for reminding us of the unclear illustrations. Since each position on the trajectory is only a camera pixel size, it is too small to see compared with the trajectory with a large overall size. Therefore, we have coloured the eight pixels around each position yellow to enlarge the visual size of each position. Moreover, we have rearranged and enlarged the overall size of Extended Fig. 5 (it is renamed as Supplementary Fig. 4 in the revised manuscript). Besides, the real trajectories have been added in the figures as red solid lines.

Action taken

The replotted trajectories are shown as below:

Supplementary Fig. 4 Recovered trajectories in the experiments. The relative positions are the white spots in the centre of the yellow squares. The red solid lines represent the real trajectories. **a**, The recovered trajectory of the object consisting of two $10\ \mu\text{m}$ width lines and moved in 22 layers of parafilm. **b**, The recovered trajectory of the object consisting of three $5\ \mu\text{m}$ width lines and moved in 14 layers of parafilm. **c**, The recovered trajectory of the object consisting of six $1\ \mu\text{m}$ width lines and moved in 6 layers of parafilm. **d**, The recovered trajectory of a number shaped '5' and moved in 14 layers of parafilm. **e**, The recovered trajectory of the fluorescent beads. Scale bars: $10\ \mu\text{m}$.

6. Reviewer comment

The choice of the name “speckle masking” isn’t obvious to me, when the operation is essentially a dot product of two high-pass filtered speckle patterns (“On the one hand, the Hadamard product is performed on speckles S_i and S_j to implement the speckle-masking operation, shown as ‘o’ in Extended Data Fig. 2; all pixel values of the masked speckle are summed to obtain a value C_{ij} of the object’s autocorrelation”)

Response

We thank the reviewer for this professional suggestion. We would like to use “overlapping speckle” instead of “speckle masking” after careful thought. Because the result of the Hadamard product between speckles only retains the overlapping speckle between them, “overlapping speckle” is clearer than “speckle masking”.

Action taken

In the revised manuscript, the sentences around Eq. (3) have been revised as follows: “Under this assumption, the overlapping speckle between these two speckles is extracted through a Hadamard product as follows:

$$\begin{aligned} M_{ij}(\xi, \eta; \Delta x_{ij}, \Delta y_{ij}) &= S_i(\xi, \eta) S_j(\xi, \eta; \Delta x_{ij}, \Delta y_{ij}) \\ &= \iint O(x, y) O(x - \Delta x_{ij}, y - \Delta y_{ij}) I_S^2(x, y) s^2(\xi, \eta; x, y) dx dy. \end{aligned} \quad (3)$$

The result only retains impulse responses from the point sources within the overlapping region of the object before and after the motion displacement $(\Delta x_{ij}, \Delta y_{ij})$.”

In addition, all the terms “speckle masking” have been changed with “overlapping speckle” throughout the manuscript.

7. Reviewer comment

The practical usefulness of this method seems to be limited by (a) the assumption of uniform object intensity, (b) the requirement for object sparsity, (c) the requirement of object stability (no transformation other than translation). In my opinion this paper is still an important and interesting contribution without solving these problems, but the authors should give the reader a detailed explanation of these limitations and illustrate where the method starts to fail.

Response

We thank the reviewer for the valuable suggestions. We have summarized the limitations of speckle kinetography as follows:

Firstly, the medium within the region of the imaging system’s field of view (FOV) should keep static during the detection. The dynamic medium will cause the impulse responses’ change so that the overlap information cannot be extracted from speckles

any more. Secondly, the illumination and medium within the region of the imaging system's FOV should be uniform. In this condition, the intensity distribution $I_S(x, y)$ within this region is nearly uniform. Otherwise, the intensity distribution of the constructed autocorrelation will be affected, which can be inferred from Eqs. (4) and (5). Thirdly, the object must be only translated but not deformed during the detection. Otherwise, the overlap information no longer represents the autocorrelation value so that the constructed object's autocorrelation will be meaningless. Fourthly, the object should be sparse. The speckle contrast decreases as the object's complexity increases since the speckle consists of more impulse responses in this case. As mentioned above, image cannot be reconstructed when the speckle contrast is extremely low. Thus, there is a limitation on the object's sparsity. Lastly, the contrast of the object to background should be high enough. If it is very low, the speckle contrast and the envelope contrast to background will be too low to achieve imaging. The type of contrast can be intensity contrast, wavelength contrast and so on.

Fig. R13 Autocorrelation construction of an object with a transmittance of gradual distribution. **a**, The transmittance distribution of a numerically simulated object. The curve represents the transmittance distribution of the object along x direction marked with a blue line. The transmittance has a gradual distribution along x direction. **b**, The correlation values along x direction of the object's autocorrelation marked with a blue line. The red solid curve represents the ground truth. The blue dashed curve is calculated from the numerically simulated speckles via speckle kinetography.

As a footnote, we described the requirement of uniform intensity distribution $I_S(x, y)$ instead of uniform object intensity. We would like to explain this briefly. From Eqs. (4) and (5), we can find that the object's autocorrelation can be constructed as long as the intensity distribution $I_S(x, y)$ is uniform. It is not affected by the distribution of the object's transmittance. Moreover, the object's transmittance distribution is contained in the constructed autocorrelation. As shown in Fig. R13a, we numerically simulated an object with a transmittance of gradual distribution. This object is numerically shifted from left to right. The speckles are numerically generated during the shift. Then, the autocorrelation values are calculated through speckle kinetography. The result is shown

in Fig. R13b, which is consistent with the ground truth although the simulated speckle contrast is not very high. This result preliminarily proves that speckle kinetography has the potential of imaging objects with uneven transmittance. Limited by time and experiment condition, experimental demonstrations are not implemented yet. We would like to do further research on this interesting topic in the future.

Action taken

We have added the above-mentioned summary about the limitations of speckle kinetography in the fourth paragraph of the Discussion in the revised manuscript.

8. Reviewer comment

Related: some sections imply a wide range of possible applications ("The motion of the object can be either spontaneous, such as blood cells under tissue, aircraft upon clouds and creatures under ice cover, or controlled, such as acoustic manipulation for cells, optical forces for particles and remote operation for drones.") which could mislead the reader as none of these examples are currently within reach of this method. Authors should either remove such examples or state explicitly that these examples are beyond the limits of the method (or provide experimental evidence to the contrary).

Response and action taken

We thank the reviewer for this helpful comment. We have removed this sentence in the Discussion and deleted similar descriptions throughout the revised manuscript.

Reviewer 3:

1. Reviewer comment

Shi et al. presented a technique termed Speckle kinetography for imaging a moving object within scattering media. The main concept involves recording the pattern of light transmitted through the scattering medium while the object moves. Information about the object's autocorrelation is obtained from the correlation between these patterns. This information is then used to retrieve the object image through a phase retrieval algorithm. The technique was demonstrated at a proof-of-concept level with samples such as vertical bars, number patterns, and a few fluorescent particles. Scattering media such as parafilm or chicken breast tissue were used, and the thickness of the scattering medium ranged up to 16.9 times the scattering mean free path. The resolution of the reconstructed object was about 1 micrometer.

While the idea itself is new, it only works under very restricted conditions. The entire

object within the scattering medium must move in a specific pattern, which is rare in practice. Furthermore, its trajectory must be precisely known to construct the object's autocorrelation map, which is said to be inferred from the background envelope. However, this is only possible when a bright object is spatially isolated. Additionally, obtaining object information from its autocorrelation map mostly applies to objects with structures as simple as those demonstrated in the study with high contrast. Typically, autocorrelation tends to reduce the contrast of an object's fine structures as spatial information about the object gets integrated, making it highly sensitive to noise. In this context, the proposed method of obtaining an object's autocorrelation from the correlation of its incoherent images appears applicable mainly to objects with high contrast. Considering these factors, the study lacks a broad impact and potential for applicability. Therefore, it would be suitable for a more specialized journal.

Response

We thank the reviewer for the thorough review and valuable comments. We agree with the reviewer that imaging of complex objects with low contrast to background is significant, which is also the ultimate goal pursued in the field of scattering imaging. Since imaging within or through strong scattering media is difficult, studies usually start with simple and high-contrast objects to explore fundamental solutions [Nature 491, 232-234 (2012)., Nat. Photonics 8, 784-790 (2014)., Nat. Commun. 13, 5779 (2022).]. On this basis, more practical solutions, such as imaging under ambient light, are studied preliminarily [Opt. Lett. 46, 4538-4541 (2021).]. However, current studies mainly focus on thin scattering layers, which have limited application scenarios. In order to broaden the application, we turn to study imaging within and through optically thick scattering media. It is more difficult than imaging through strong scattering layers because little optical information survives after multiple scattering. In this case, we choose simple and high-contrast objects to start our research.

Besides, we studied the influence of the object's contrast to background on speckle kinetography through a group of simulation experiments (also described below in detail). The results show that imaging can be achieved not only when the contrast is 1/0, but also when it is higher than 1:0.5. Therefore, object's high contrast to background is not strictly required for imaging, thereby affording us potential avenues for further enhancements in speckle kinetography. Moreover, beyond intensity contrast, the utilization of wavelength contrast, exemplified by techniques such as fluorescence imaging and infrared thermal imaging, ensures the applicability of speckle kinetography across a specific range of applications.

Fig. R14 Comparison of scattering imaging with different motion trajectories. **a**, Motion trajectories required by the existing methods²⁵⁻²⁷. The trajectories must cover at least half of the autocorrelation. **b**, Object's autocorrelation constructed from the recorded speckles generated from the object moved with trajectories in (**a**). **c**, Object image recovered from (**b**). **d**, A simplified motion trajectory used by speckle kinetography. **e**, Object's autocorrelation constructed from the recorded speckles in (**d**). **f**, Object image recovered from (**e**). **g**, The recovered trajectory from an unknown and uncontrolled object. The blue dotted lines in T shape can be selected for imaging. **h**, Object's autocorrelation constructed from the recorded speckles in the T-shaped trajectory of (**g**). **i**, Object image recovered from (**h**).

Since imaging in optically thick scattering media without prior knowledge is very difficult, some changes, such as relative motions or illumination variations, seem to be necessary at present to compensate the destructed spatial invariance. Under this condition, speckle kinetography has minimized the requirement of motion trajectory compared with the existing methods²⁵⁻²⁷. For example, motion trajectories with previously known positions covering at least half of the autocorrelation (Fig. R14a) are required for object's autocorrelation construction (Fig. R14b) and image reconstruction (Fig. R14c) at present²⁵⁻²⁷, which is hard to implement in practical applications. Through speckle kinetography, a simplified trajectory (Fig. R14d) can also provide the object's autocorrelation of the same size and resolution (Fig. R14e) as well as the reconstructed image of close quality (Fig. R14f). Moreover, the positions in the trajectory are not previously known but recovered from the cross-correlations between every two envelopes filtered from the recorded speckle images. Notably, the object does not have to move in a specific pattern. For example, here is an unknown and uncontrolled object. We record as many speckle images as possible at the shortest

possible shooting intervals during the object motion. According to the positions extracted from the envelopes, a trajectory is recovered (Fig. R14g). Then, a portion of the trajectory, which has side lengths larger than the speckle pattern, is selected for computational processing (blue dotted T shape in Fig. R14g). As shown in Figs. R14h and R14i, the autocorrelation of this T-shaped trajectory is large enough to sample the whole object's autocorrelation for imaging. If the computational capacity is sufficient, even the whole trajectory shown in Fig. R14g can be utilized for imaging without selection. Therefore, the only requirement is recording speckle images as many and fast as possible during the object motion, regardless of the object's shape or motion trajectory. With the remarkable reduction in trajectory requirements, speckle kinetography presents new and promising opportunities to address certain limitations encountered in practical applications.

Given the progress achieved in deep imaging depth and high resolution, coupled with the considerably diminished prerequisites for trajectory and illumination, we believe speckle kinetography holds immense potential to make substantial contributions to the field of scattering imaging that continues to merit consideration for publication in *Nature Communications*.

Action taken

In the revised manuscript, we have added the descriptions about the limitations of speckle kinetography in the Discussion, including the requirements for the object's high contrast and sparsity to make the limitations clear to the reader. The related descriptions about the limitations are as follows:

“Fourthly, the object should be sparse. The speckle contrast decreases as the object's complexity increases since the speckle consists of more impulse responses in this case. As mentioned above, image cannot be reconstructed when the speckle contrast is very low. Thus, there is a limitation on the object's sparsity. Lastly, the contrast of the object to background should be high enough. If it is very low, the speckle contrast and the envelope contrast to background will be too low to achieve imaging. The type of the contrast can be intensity contrast, wavelength contrast and so on.”

The trajectory-related descriptions have also been rewritten in the Results and the Methods of the manuscript to make it clear to the readers.

2. Reviewer comment

Here are a few comments that need consideration:

The layout of Fig. 1 seems ambiguous to properly convey the concept. It would be better to arrange the subpanels in a way that directly represents Equations 1 and 2.

Response and action taken

We thank the reviewer for this helpful suggestion. In the revised manuscript, the insets in Fig. 1 have been rearranged to directly represent Eqs. 1 and 2. The modified illustration becomes clear and visualized under the reviewer's professional guidance, which is shown as follows:

Fig. 1 Schematic of the principle, apparatus and computational model for speckle kinetography. **a**, Principle. Under incoherent illumination, the intensity distribution on object plane is $I_S(x, y)$. Every point source forms an impulse response $s(\xi, \eta; x, y)$ on the back surface of scattering medium. The recorded speckle image I is the superposition of the impulse responses generated within the object. The information of overlap C and relative position $(\Delta x, \Delta y)$ between any two motion states of an object is retained in the high-pass filtered speckles S and low-pass filtered envelopes E . The information is fully utilized to construct the object's autocorrelation $C(\Delta x, \Delta y)$ for imaging. **b**, Experimental setup. A movable object is embedded in an unknown scattering medium, which can be optically thick scattering media or thin scattering layers. A simple and speckle pattern-sized trajectory is sufficient for imaging. The front surface of the medium is illuminated via an incoherent light source, which can either be narrowband or broadband, and speckles are formed at the back surface of the medium. A series of magnified speckle images I are noninvasively detected via a simple imaging system consisting of a lens and a monochrome camera. **c**, Computational model. The speckle images are numerically separated into spatially normalized speckles S and slowly varying envelopes E . The overlap C is calculated from the sum of the overlapping speckle M , which is sifted from any two of the speckles by Hadamard product. The relative position $(\Delta x, \Delta y)$ is calculated from the corresponding two envelopes via cross-correlation. Combining all the extracted information, the object's autocorrelation is constructed and then utilized for object image reconstruction.

3. Reviewer comment

In actual measurements, what is the ratio of the envelope to the object's autocorrelation component? And what determines this?

Response

We thank the reviewer for pointing out this concern. The construction of object's

autocorrelation consists of two main parts. One part is to determine the sampling area of the object's autocorrelation according to the trajectory, which is recovered from the envelopes. The other part is to determine the autocorrelation values of every pixel on the sampling area according to the speckles. In that sense, the envelope contributes half to the construction of object's autocorrelation.

Fig. R15 The contribution of the envelope to object's autocorrelation construction in actual measurements. **a**, An object consisting of three 25 μm width lines. It is embedded in the parafilm. **b**, One of the recorded speckle images. **c**, The low-pass filtered envelope. **d**, The speckle obtained by dividing the speckle image with its envelope. **e**, The object positions extracted from cross-correlation of the envelopes. **f**, The calculated motion trajectory. **g**, The sampling area calculated from the autocorrelation of the trajectory in (f). **h**, The object's autocorrelation constructed by placing the values calculated from the speckles into the sampling area in (g). Scale bars: 50 μm .

To explain in detail, we would like to describe the envelope-based object's autocorrelation construction in actual measurements in the following. As shown in Fig. R15a, an object consisting of three 25 μm width lines is embedded in the parafilm. One of the recorded speckle images is shown in Fig. R15b. The envelope (Fig. R15c) is low-pass filtered from the speckle image, and then the speckle (Fig. R15d) is obtained by dividing the speckle image with its envelope (Supplementary Note 11). Then, the cross-correlation of the envelope and another envelope is calculated, and the position of its maximum value is extracted as a position in the trajectory. In the same way, all the positions in the trajectory are calculated and shown as the points in Fig. R15e, which consist a trajectory. According to the average distance between adjacent two positions, the pixel size of the object's autocorrelation is determined, which corresponds to 6 camera pixels in this experiment. Although the calculated positions have some deviations from the real trajectory (res solid line in Fig. R15e), but the deviations are within 2 camera pixels. That is, the deviations are smaller than half pixel size of the object's autocorrelation. Therefore, the recovered trajectory shown in Fig. R15f is not deformed. On this basis, the sampling area is determined from the autocorrelation of

the calculated trajectory, shown as the square area with positive values in Fig. R15g. By placing the autocorrelation values calculated from the spatially normalized speckles into the corresponding positions in the sampling area, the object's autocorrelation is constructed (Fig. R15h).

Since the envelopes only determine the sampling positions for object's autocorrelation component rather than the values of the autocorrelation component, the intensity ratio of the envelope to the object's autocorrelation component is meaningless. Nevertheless, the contribution ratio of the envelope to the object's autocorrelation component construction can be regarded as 0.5.

Action taken

In order to give the readers a clear explanation about the contribution of the envelope to object's autocorrelation construction, the method for autocorrelation construction with a simple trajectory extracted from the envelopes has been rewritten in the Methods of the manuscript as follows:

“Autocorrelation construction with a simple trajectory

Under incoherent illumination, the object's autocorrelation can be constructed with a simple trajectory with side lengths slightly larger than the speckle pattern. For example, speckle images generated from a moving object are recorded. According to the cross-correlations between every two envelopes filtered from the speckle images, all the relative positions of the object are extracted, which consist the recovered trajectory of the object (Supplementary Fig. 3a). The side lengths of the trajectory are slightly larger than the overall size of the speckle pattern, which is larger than the object size due to scattering. The autocorrelation of this U-shaped trajectory is shown in Supplementary Fig. 3b. The value of each position on this autocorrelation reflects the number of the corresponding relative position $(\Delta x_{ij}, \Delta y_{ij})$ contained in this trajectory. Therefore, according to Eq. (5), the object's autocorrelation can be sampled at the positive-valued positions of the trajectory's autocorrelation. As shown in Supplementary Fig. 3c, in the positive-valued area of the trajectory's autocorrelation, the object's autocorrelation is completely sampled. The value of each pixel in the sampling area are calculated from the corresponding spatially normalized speckles according to Eq. (5). Besides, as the object's line width is not known, the interval between every two shots is set as small as possible to ensure the resolution for imaging (Supplementary Fig. 3d).

In actual detections, the object is unknown and uncontrolled. In this case, we record as many speckle images as possible at the shortest possible shooting intervals. According to the cross-correlations between every two envelopes, the trajectory is recovered (Supplementary Fig. 3e). Then, we select a T-shaped portion of the trajectory

for computational processing (blue dotted T shape in Supplementary Fig. 3e) since its side lengths are slightly larger than the speckle pattern. As shown in Supplementary Figs. 3f and 3g, the autocorrelation of this T-shaped trajectory is large enough to cover and sample the entire autocorrelation of the object. Based on this, the object image is reconstructed (Supplementary Fig. 3h). Besides, since the motion interval of spontaneously moving objects cannot be fully controlled, the imaging resolution requirements may not be met in some extreme cases.

The U and T shaped trajectories are just two workable examples. In fact, no matter what shape the trajectory is, as long as its autocorrelation can cover the entire autocorrelation of the object with sufficient resolution, the object image can be reconstructed. If the computational capacity is sufficient, the whole trajectory shown in Supplementary Fig. 3e can be utilized for imaging without selection. Therefore, the only requirement is recording speckle images as many and fast as possible during the object motion, regardless of the object's shape or motion trajectory.

Supplementary Fig. 3 Autocorrelation construction with a simple trajectory. The object image can be reconstructed as long as the autocorrelation of the trajectory is large enough to cover the object's autocorrelation with sufficient resolution. **a**, Speckle image from a hidden object. The trajectory extracted from the cross-correlations between every two envelopes is in a U shape. The overall size of the trajectory is slightly larger than the speckle pattern. **b**, Autocorrelation of the U-shaped trajectory in **(a)**. **c**, The constructed autocorrelation of the object with a sampling area determined by **(b)**. **d**, The object image recovered from **(c)**. **e**, Speckle image from a spontaneously moving object. The trajectory is recovered from the cross-correlations between every two envelopes. **f**, Autocorrelation of the partially selected trajectories shown as a blue dotted T shape in **(e)**. **g**, The object's autocorrelation with a sampling area determined by **(f)**. **h**, The object image recovered from **(g)**. Scale bars: 50 μm ."

4. Reviewer comment

The position of the object is inferred from the envelope, a non-interfering component. This method ultimately relies on detecting the displacement of the envelope due to the object's movement. This process seems to require an object to be isolated. Additionally, if the envelope is too broad, the precision of detecting movement could decrease.

Response

We thank the reviewer for pointing out this concern and would like to explore the influence of the intensity contrast between background and object on imaging through a group of simulation experiments.

Fig. R16 Influence of the contrast of background to object, C_{bo} , on speckle kinetography by simulation experiments. a, The relative displacement error E versus C_{bo} . **b,** The SSIM of the constructed object's autocorrelation versus C_{bo} . **c,** The SSIM of the reconstructed object image versus C_{bo} .

The hidden object consists of three lines with an intensity transmittance of 1. The intensity transmittance of the background varies from 0 to 0.9, in steps of 0.1. Therefore, the intensity contrast of the background to the object, C_{bo} , varies from 0/1 to 0.9/1. The simulated scattering properties corresponds to parafilm of about 14 layers. The dynamic range of the simulated camera is 8 bits referring to the camera we used in experiments.

The relative displacements are recovered through cross-correlations of the low-pass filtered envelopes. The relative displacement error E fluctuates within ± 3 camera pixels as C_{bo} increases (Fig. R16a). But the fluctuation does not cause distortion of the autocorrelation since the pixel size of the constructed object's autocorrelation corresponds to 10 camera pixels in this simulation. When the sample becomes thicker, the camera should be changed to the one with a larger dynamic range. Otherwise, the envelope's contrast to background would be too low to be distinguished by the camera.

The object's autocorrelation is constructed by speckle kinetography. The structural similarity index measurement (SSIM) of the constructed autocorrelation decreases with C_{bo} (Fig. R16b). When C_{bo} is bigger than 0.5, the SSIM decreases rapidly because the speckle contrast becomes low owing to the enhanced background light. The SSIM of the reconstructed object image also decreases with C_{bo} and is acceptable when C_{bo} is within 0.5 (Fig. R16c).

Under the same conditions, when C_{bo} is bigger than 0.5, the relative displacement error keeps within the fault tolerance but the autocorrelation value distributions are already wrong. Therefore, the failure of imaging is primarily caused by the low speckle contrast rather than the low envelope contrast to background. In addition, the high

object's contrast to background is not a very strict requirement according to the simulation results.

Fig. R17 Influence of envelope broadening on relative displacement error. Experiments of an object consisting of three 25 μm width lines hidden inside parafilm of 2 to 30 layers, in steps of 2 layers. **a**, The FWHM W of the envelopes versus the sample thickness L . **b**, The relative displacement error E versus L .

To further study the influence of envelope broadening on the relative displacement error, we implemented a group of experiments by increasing the sample thickness. In these experiments, the scattering samples were parafilm of 2 to 30 layers, in steps of 2 layers. A transmissive object consisting of three 25 μm width lines was embedded in the middle of the parafilm. As shown in Fig. R17a, the full width at half-maximum (FWHM) W of the envelope broadens as the sample thickness L increases. The curve in Fig. R17b shows that the relative displacement error E fluctuates slightly as L increases. And the fluctuation keeps within ± 4 camera pixels. In this experiment condition, the pixel size of the constructed object's autocorrelation corresponds to 16 camera pixels. Therefore, fluctuations within ± 8 camera pixels do not cause misplacement to autocorrelation pixels. Since the speckle contrast is too low to achieve imaging when the sample thickness is larger than 26 layers, the relative displacement error is not significantly affected by envelope broadening within the imageable range.

As a conclusion, the speckle contrast, the envelope broadening and the envelope contrast to background limit the imageable object's contrast, and the speckle contrast dominates. To achieve high speckle contrast, high object's contrast to background is a basic requirement, but it is not strict according to the simulation and experiment results, thereby affording us potential avenues for further enhancements in speckle kinetography.

Action taken

In the revised manuscript, the discussions about the envelope broadening and relative displacement error have been added in the Discussion as follows:

“The full width at half-maximum W of the envelope broadens as the sample thickness L increases (Supplementary Fig. 6b). But the entire envelope can still be measured when the parafilm is 28 layers, so it does not have obvious impact on imaging. The relative displacement error E fluctuates slightly as L increases (Supplementary Fig. 6c). But the fluctuation keeps within ± 4 camera pixels, which is smaller than the pixel size of the object’s autocorrelation, corresponding to 16 camera pixels in these experiments. Thus, it does not cause misplacement to autocorrelation pixels.” The Supplementary Figs. 6b and 6c mentioned in this paragraph is the same as Fig. R17.

5. Reviewer comment

The resolution of image reconstruction appears to be determined by the step of the object's movement, and the resolution and field of view of the imaging system. A theoretical model and analysis are needed.

Response

We thank the reviewer for this valuable suggestion and would like to make a theoretical analysis of the imaging resolution.

The following conditions are required for sharp imaging of objects via speckle kinetography. Firstly, the field of view (FOV) of the imaging system can cover the entire speckles patterns in the trajectory. Secondly, the resolution of the imaging system should be smaller than the speckle grain size. Thirdly, the interval distance of the moving object should be smaller than the narrowest line width of the object. Under the above conditions, the imaging resolution R_{SK} of speckle kinetography is determined as follows:

$$R_{SK} = \begin{cases} R_{obj}(R_{IS}, L) & \text{for } R_{obj} > T_{obj} \\ T_{obj} & \text{for } R_{obj} \leq T_{obj} \end{cases},$$

where T_{obj} represents the interval distance of the moving object and R_{obj} represents the resolution limit on the object plane. In scattering imaging, the resolution limit R_{IS} of the imaging system determines the minimum detectable speckle grain size. Since the speckle grain size D increases with the object’s line width LW (Fig. R18a), the minimum detectable speckle grain size determines the resolution limit R_{obj} on the object plane. Therefore, R_{IS} determines R_{obj} . In addition, the speckle grain size D decreases as the medium thickness L increases (Fig. R18b), that is, the same sized speckle scattered from the thicker sample should be generated from objects with the larger line width. Therefore, the resolution limit R_{obj} on the object plane becomes larger when the medium thickness L increases. Therefore, L also affects R_{obj} . In conclusion, R_{obj} is determined by R_{IS} and L .

According to the above equation, the resolution limit of speckle kinetography depends on the resolution limit R_{obj} on the object plane and the minimum interval distance of the object. We assume that the resolution limit R_{IS} of the imaging system reaches the optical diffraction limit. Since R_{obj} becomes larger as L increases, R_{obj} can only be close to rather than equal to the diffraction limit R_{IS} . Besides, the minimum interval distance of the object must be longer than the coherence length. As the interval distances in the experiments for 1 μm widths objects were already shorter than the optical diffraction limit, the minimum interval distance is not the main limit compared with R_{obj} . In conclusion, the resolution limit of speckle kinetography is theoretically close to and slightly larger than the optical diffraction limit.

Fig. R18 The speckle grain size versus the object size and the medium thickness. a, The average speckle grain size D increases with the object's line width LW . The scattering sample is parafilm of 14 layers. The objects consist of three lines with 5, 10, 16, 20 and 25 μm widths, respectively. **b**, The average speckle grain size D decreases as the medium thickness L increases. The scattering samples are parafilm of 2 to 30 layers, in steps of 2 layers. A transmissive object consisting of three 25 μm width lines is embedded in the middle of the parafilm.

Action taken

In the revised manuscript, the theoretical analysis of the imaging resolution described above has been added in the third paragraph of the Discussion.

6. Reviewer comment

Parafilm and chicken breast tissue were used as the scattering media, but what is the distance between the object and the scattering medium? In fact, if the distance is large, high-angle multiple scattering can effectively be filtered out, which leads to an overestimation of the effect of the scattering medium. Typically, objects are not separated from the scattering medium, but embedded within it.

Response

We thank the reviewer for pointing out this ambiguity. The objects were embedded within the scattering samples in all experiments. The distances between the object and the scattering medium were near zero.

Action taken

In the revised manuscript, the information of the distance between the object and the scattering medium has been revised in the ‘Overlapping speckle correlations’ section of the Results and the ‘Experimental setup’ section of the Methods as follows:

“As shown in Fig. 1b, an object with an intensity transmittance of $O(x, y)$ is embedded in a scattering medium that is illuminated via an incoherent light source.”

“The scattering samples had gaps in their middle sections that were approximately equal to the thickness of the object and were held by a plate holder.”

“The object was embedded in the scattering samples and controlled via a two-axis motorized precision translation stage (100 nm minimum step size along the horizontal and vertical directions).”

REVIEWERS' COMMENTS

Reviewer #1 (Remarks to the Author):

The authors have addressed the raised concerns by conducting additional experiments and simulations. The revised manuscript offers a detailed explanation of the method's principles, and conditions/limitations regarding trajectory and depth limit.

One minor comment is that the light source seems not explained for the experiment in supplementary note 6 (imaging depth limit).

Reviewer #2 (Remarks to the Author):

The authors answered my comments in detail and addressed my concerns. I appreciate the expanded discussion pointing out the various practical limitations.

One last point: I find it highly unusual that they added the description of their new results to the beginning of the Discussion, rather than to the Results or Supplement (I recommend the latter). Assuming that this will be fixed, I support publication.

Reviewer #3 (Remarks to the Author):

In view of the revised manuscript by Shi et al., I acknowledge that the authors have made a faithful effort to address my questions and concerns. However, the limitations of the method presented in this study remain unchanged. Specifically, it is only applicable to extremely simple resolution targets or a few fluorescent particles. Additionally, the target object must have sufficiently high contrast, and its movement distance must be considerable. These conditions are rarely met in real-world scenarios. While one can often accommodate these kinds of limitations in earlier pioneering studies due to the innovative nature of the work, I find that the novelty of the concepts introduced in this study does not justify these constraints. Consequently, I maintain that this paper would be better suited for a more specialized journal.

Response to reviewers

We appreciate it very much for the reviewers' valuable suggestions and comments that greatly improve our paper. We have addressed the reviewers' comments point by point and prepared a revised manuscript accordingly. Following this letter are our responses to the reviewers' comments.

Reviewer 1:

Reviewer comment

The authors have addressed the raised concerns by conducting additional experiments and simulations. The revised manuscript offers a detailed explanation of the method's principles, and conditions/limitations regarding trajectory and depth limit.

One minor comment is that the light source seems not explained for the experiment in supplementary note 6 (imaging depth limit).

Response:

We are sincerely grateful for the reviewer's careful review and helpful advices. The narrowband and broadband light sources used in the experiments in Supplementary Note 6 are respectively a single-colour LED (625 nm nominal wavelength, 17 nm bandwidth, 920 mW) and a research arc lamp source (260 to 2500 nm wavelength, 150 W). We have added the explanation about the light sources in the second paragraph of Supplementary Note 6.

Reviewer 2:

Reviewer comment

The authors answered my comments in detail and addressed my concerns. I appreciate the expanded discussion pointing out the various practical limitations.

One last point: I find it highly unusual that they added the description of their new results to the beginning of the Discussion, rather than to the Results or Supplement (I recommend the latter). Assuming that this will be fixed, I support publication.

Response:

We thank the reviewer for the kind suggestions and valuable comments. The experimental results about the imaging depth limit described in the first two paragraphs of the Discussion have been moved into Supplementary Note 6. And the first two

paragraphs of the Discussion have been revised as follows:

“As the thickness of scattering sample increases, the speckle contrast decreases. Meanwhile, the envelope gradually broadens so that the relative displacement of the envelope will not be accurately extracted especially when the entire envelope cannot be measured. These factors limit the imaging depth. Nevertheless, when the speckle contrast becomes too low to realize imaging, the entire envelope can still be measured and the relative displacement is still accurate enough for achieving the correct autocorrelation construction (see Supplementary Note 6 for experimental demonstrations). Therefore, the speckle contrast is a critical factor for the imaging depth. Under narrowband and broadband illumination, the experimental results of imaging in anisotropic and isotropic scattering samples, which respectively are parafilm and polyethylene foams²⁴, show that the maximum imaging depth of speckle kinetography is about $6l_t$ (Supplementary Note 6).”

In addition, the seven sentences describing the experimental results about the imaging resolution in the third paragraph of the Discussion has been moved into Supplementary Note 7. Instead, a summary sentence has been added here as follows:

“The R_{obj} is determined by the resolution limit of the imaging system R_{IS} and the sample thickness L (Supplementary Note 7).”

We are sincerely grateful for the reviewer’s helpful suggestions that greatly improve our paper.

Reviewer 3:

Reviewer comment

In view of the revised manuscript by Shi et al., I acknowledge that the authors have made a faithful effort to address my questions and concerns. However, the limitations of the method presented in this study remain unchanged. Specifically, it is only applicable to extremely simple resolution targets or a few fluorescent particles. Additionally, the target object must have sufficiently high contrast, and its movement distance must be considerable. These conditions are rarely met in real-world scenarios. While one can often accommodate these kinds of limitations in earlier pioneering studies due to the innovative nature of the work, I find that the novelty of the concepts introduced in this study does not justify these constraints. Consequently, I maintain that this paper would be better suited for a more specialized journal.

Response:

We thank the reviewer for the valuable suggestions and comments that greatly improve

our paper. Since imaging through strong scattering media is greatly challenging, some limitations remain in this field so far, such as imaging depth, being invasive, imaging resolution, stationary object, illumination, object complexity, high contrast, and so on [Nat. Commun. 13, 1447 (2022)., Nat. Commun. 13, 4081 (2022)., Nat. Commun. 13, 5779 (2022)., Nat. Rev. Phys. 2, 141-158 (2020).]. In this work, we have considerably extended the incoherent imaging depth range for moving object while achieving high resolution and being noninvasive. In addition, an object-sized movement distance is enough for imaging. The high contrast can be achieved through not only intensity but also wavelength, exemplified by techniques such as fluorescence imaging and infrared thermal imaging, ensures the adaptability of speckle kinetography across a specific range of applications. Despite the common challenge of object complexity in speckle-based imaging techniques [Nat. Commun. 13, 1447 (2022)., Nat. Commun. 13, 4081 (2022)., Nat. Commun. 13, 5779 (2022)., Nat. Photonics 8, 784-790 (2014)., Nature 491, 232-234 (2012).], we are dedicated to ongoing efforts aimed at identifying strategies to surmount this obstacle. With the significant advancements made in deep imaging depth and high resolution, alongside the substantially reduced requirements for trajectory, illumination, and prior knowledge, we believe that speckle kinetography possesses enormous potential to make substantial contributions to the field of scattering imaging.

We sincerely appreciate the valuable insights provided by the reviewer regarding the limitations of our method. These comments have served as a catalyst for us to explore novel approaches in order to address these challenges. Furthermore, we are committed to further enhancing and expanding the scope of our method, which holds promising potential for significant applications in specialized scenarios.